# MoTE: Reconciling Generalization with Specialization for Visual-Language to Video Knowledge Transfer

**Minghao Zhu**     **Zhengpu Wang**     **Mengxian Hu**     **Ronghao Dang**
**Xiao Lin**     **Xun Zhou**     **Chengju Liu**[*]     **Qijun Chen**

Tongji University, Shanghai, China

{zmhh_h, wangzhengpu, humengxian, dangronghao, linx_xx,
zhouxun, liuchengju, qjchen}@tongji.edu.cn

## Abstract

Transferring visual-language knowledge from large-scale foundation models for video recognition has proved to be effective. To bridge the domain gap, additional parametric modules are added to capture the temporal information. However, zero-shot generalization diminishes with the increase in the number of specialized parameters, making existing works a trade-off between zero-shot and close-set performance. In this paper, we present MoTE, a novel framework that enables generalization and specialization to be balanced in one unified model. Our approach tunes a mixture of temporal experts to learn multiple task views with various degrees of data fitting. To maximally preserve the knowledge of each expert, we propose *Weight Merging Regularization*, which regularizes the merging process of experts in weight space. Additionally with temporal feature modulation to regularize the contribution of temporal feature during test. We achieve a sound balance between zero-shot and close-set video recognition tasks and obtain state-of-the-art or competitive results on various datasets, including Kinetics-400 & 600, UCF, and HMDB. Code is available at https://github.com/ZMHH-H/MoTE.

## 1 Introduction

With the advent of large-scale Vision-Language Models (VLMs), adapting such foundation models (e.g., CLIP [33], ALIGN [13], Florence [54]) for downstream tasks has become an emerging paradigm of intense scrutiny. Their semantic visual concepts empowered by aligning large-scale image-text pairs can be transferred to a wide range of downstream tasks, such as open-vocabulary classification [55, 56], detection [11, 51], and segmentation [21, 52].

The crux of an adaptation procedure for foundation models lies in injecting specialized knowledge of the domain of interest. For video recognition tasks, this necessity is reflected in the fact that the dynamic nature of video data requires effective comprehension of context and temporal correlation by the model. Therefore, to condition the adapted model on the video-specialized knowledge, one general and effective way is to incorporate additional parameters in the form of well-designed prompts [43, 45], adapters [30, 53], and temporal modules [28, 49, 50]. However, we observe that while the increased model capacity brought by more additional parameters enables the better fitting of video-specific inductive bias, it comes at the cost of catastrophically forgetting the generalization knowledge of the original VLM. To better illustrate this phenomenon, we present an overview of existing methods in Figure 1. A clear trade-off problem between zero-shot and close-set performance emerges in existing works, which correlates to the scale of additional parameters. The former relies more on the generalization capability inherent in VLMs while the latter requires intensive video-specialized knowledge, but no method achieves the best of both worlds. On one hand, introducing new knowledge

---

[*]Corresponding author.

38th Conference on Neural Information Processing Systems (NeurIPS 2024).

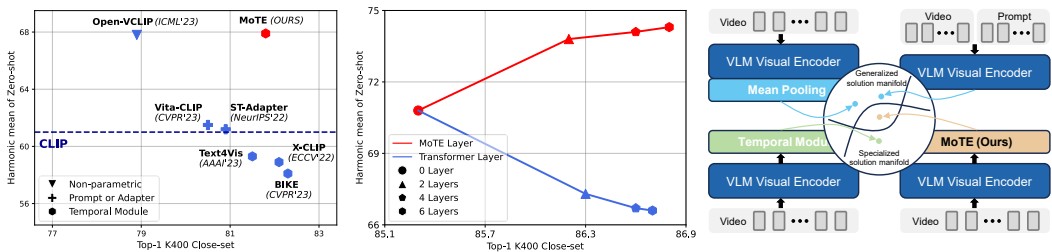

(a) Zero-shot vs. Close-set performance (b) Performance trade-off of temporal layer numbers. (c) Feature space of various VLM transfer methods

Figure 1: Overview of existing VLM knowledge transfer methods. (a) Trade-off plots between zero-shot (Harmonic mean of UCF, HMDB, and K600) and close-set (K400) performance of recent CLIP-based methods (ViT-B/16). (b) As the number of temporal layers increases, the generalization of the standard Transformer layer severely degrades while our proposed MoTE consistently improves the zero-shot and close-set performance. (c) Our proposed MoTE seeks to construct a reconciled feature space between the optimal generalized and specialized manifolds.

while preserving existing knowledge continually is desirable and significant for the broader adaptation of the foundation model. The rapidly evolving real-world applications also require both specialization and generalization capabilities. However, how to manage the generalization/specialization trade-off of additional parameters in transfer learning remains under-explored.

This paper addresses the aforementioned challenge by delving into two inquiries: (**i**) *How can the generalization capability of additional parameters be enhanced?* Given the substantially smaller scale of the fine-tuning dataset compared to the pre-training dataset of VLMs, the newly added parameters risk overfitting the fine-tuning data bias, thereby constraining the model generalization. Addressing this question enables steering parameters towards high generalization. Nevertheless, generalization and specialization have proven to be somewhat conflicting [18, 29, 36] in model training, prompting us to further explore (**ii**) *How can generalization and specialization coexist in one unified model?* This question is crucial for a balanced model to preserve both new and existing knowledge, yet has received limited exploration. We investigate these two questions with a widely employed temporal module [14, 49, 50] (i.e. N-layer transformer), which already features a high specialization degree but is less performant on zero-shot tasks.

With this in mind, we present MoTE, a mixture-of-temporal-experts approach with well-balanced generalization and specialization. We offer a novel perspective in addressing the first question: constructing a more generalized model using multiple data bias views. In contrast to conventional temporal modules that encode patterns with a *single* feedforward network (FFN) in each Transformer layer, MoTE uses *multiple* FFN experts to capture various data bias views. During fine-tuning, incoming frame token sequences are routed to one of the temporal experts, keeping the computational cost the same as using a single FFN. To enlarge the discrepancy in knowledge learned by each expert, we devise a routing algorithm based on the multinomial distribution. During inference, we apply weights merging to collapse multiple experts into one module, enabling the patched model to aggregate the generalized knowledge of each expert. For the second question, we propose *Weight Merging Regularization* which regularizes the merging process of experts in the weight space. The proposed regularizer drives a range of the merged parameters optimal with respect to the task-specific objective, allowing a more effective aggregation of generalized and specialized knowledge in the patched model. To further alleviate the overfitting at test time, we devise a plug-and-play module that modulates the contribution of temporal features by measuring the semantic association between the proxy text features from the fine-tuning and the test datasets.

Our main contributions can be summarized as follows:

- We propose MoTE, a knowledge transfer framework from visual-language to video domain. MoTE tackles the fitting trade-off challenge posed by additional parameters, an aspect largely overlooked by previous works.
- We provide new insights for enhancing parameter generalization from the perspective of data bias fitting, all while keeping the computation cost and final structure constant (§ 3.1, § 3.2).
- We propose Weight Merging Regularization, a novel regularizer for more effective knowledge aggregation, realizing the coexistence of generalization and specialization in weight space (§ 3.3). Together with Temporal Feature Modulation to further improve the generalization of MoTE (§ 3.4).

- Extensive experiments demonstrate MoTE achieves an optimal trade-off between zero-shot and close-set performance with one unified model. Thorough ablation studies show the scalability and effectiveness of our proposed method (§ 4).

## 2 Preliminary: Transferring CLIP for Video Recognition

Recent works have adapted CLIP [33] to video datasets and obtained superior results [28, 41]. We briefly describe the typical cross-modal video recognition pipeline. Consider a video $V$ with $T$ frames and the corresponding text label $C$ described in a set of textual prompts. Each frame is encoded independently by the CLIP visual encoder $f(\cdot|\theta_v)$ with parameter $\theta_v$ and produces frame-level embeddings $\{\mathbf{e}_i \in \mathbb{R}^D\}_{i=1}^T$. The text embedding $\mathbf{y} \in \mathbb{R}^D$ is generated by the CLIP text encoder $g(\cdot|\theta_c)$ with parameter $\theta_c$, where $D$ is the embedding dimension,

$$\mathbf{e}_1, \cdots, \mathbf{e}_T = f(V|\theta_v), \ \ \mathbf{y} = g(C|\theta_c). \tag{1}$$

The CLIP visual encoder captures rich spatial information. To bridge the domain gap between image and video, we apply a commonly used temporal module $h(\cdot|\theta_{tem})$ for cross-frame communication [14, 49, 50], which is parameterized by several Transformer layers. The final video embedding $\mathbf{z} \in \mathbb{R}^D$ consists of spatial and temporal embeddings connected in a residual form:

$$\mathbf{z} = \mathrm{AvgPool}([\mathbf{e}_1, \cdots, \mathbf{e}_T] + h(\mathbf{e}_1, \cdots, \mathbf{e}_T|\theta_{tem})). \tag{2}$$

During optimization, the text encoder is typically frozen. We tune the visual encoder and the temporal module to maximize the similarity $\mathrm{sim}(\mathbf{z}, \mathbf{y}) = \frac{\langle \mathbf{z}, \mathbf{y} \rangle}{\|\mathbf{z}\|\|\mathbf{y}\|}$ between the video embedding $\mathbf{z}$ and the text embedding $\mathbf{y}$ if $V$ and $C$ are matched, otherwise minimize it. The training objective can be formulated as:

$$\mathcal{L}(\theta_v, \theta_{tem}; \mathcal{D}) = \mathbb{E}_{(V,C)\sim\mathcal{D}}[\mathcal{I}(\mathrm{sim}(\mathbf{z}, \mathbf{y}), \mathrm{onehot}(C))], \tag{3}$$

where $\mathcal{D}$ is the fine-tuning dataset, $\mathcal{I}$ is the cross-entropy function with softmax operation.

## 3 Methodology

An overview of our proposed method is presented in Figure 2. This section first analyzes the parameter generalization from the perspective of data bias fitting (§ 3.1). Then we detail the structure, routing policy, and inference of MoTE (§ 3.2). Finally, we present Weight Merging Regularization (§ 3.3) and Temporal Feature Modulation (§ 3.4) towards the coexistence of generalization and specialization.

### 3.1 Intuition and Motivation

Typically, more additional parameters allow for better fitting of the training data distribution, leading to better close-set performance. However, steering the model specialized on the target distribution potentially makes it sensitive to out-of-distribution shifts, resulting in the generalization drop on downstream data distributions with unseen video categories. Although one could enlarge the training data distribution to cover as many potential unseen categories as possible, the costly computation and collection of video data make this infeasible, indicating the need for a more generalized architecture.

Neural network optimization has many solutions in different loss basins due to the non-convexity of the loss landscape. Models optimized with different configurations (e.g. initialization, optimizer, and data) have different optimization trajectories and may converge to separate local minima, thereby capturing various feature patterns. This inspires our method to construct a more generalized model using multiple data bias views. Instead of improving generalization by searching for a flatter minimum in the loss landscape [22, 48], we expect the aggregation of diverse knowledge from multiple minima can provide a more comprehensive representation. Intuitively, diverse temporal patterns can better facilitate recognizing an unseen video category. For example, aggregating information on 'player movement' and 'racket-ball interaction' can help better recognize 'playing tennis'. Our architecture design takes inspiration from Mixture-of-Experts [37] to learn multiple data bias views with a set of experts. While MoE was originally proposed for building large pre-trained models, we extend it to the context of transfer learning and demonstrate its effectiveness in improving parameter generalization.

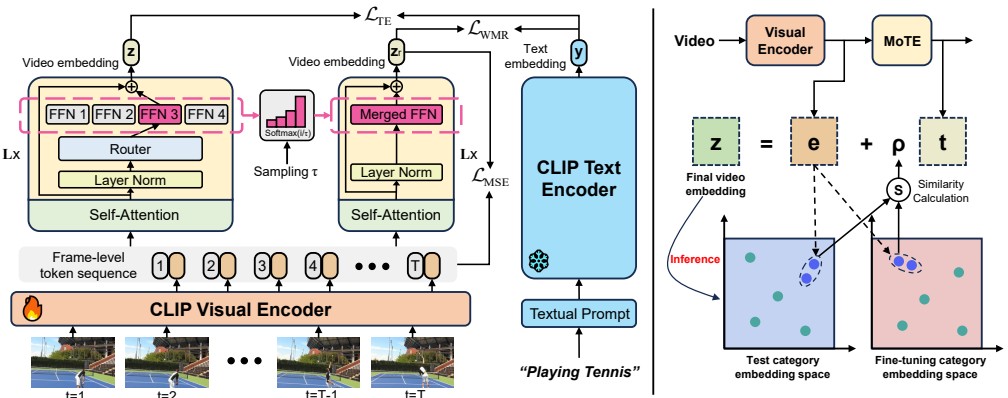

Figure 2: An overview of the MoTE framework. **(Left):** We independently extract the feature of each frame with the CLIP visual encoder. Then, the frame token sequences from a given batch are routed to an activated expert for temporal pattern encoding. To regularize the merging process, we sample the temperature $\tau$ from a discrete set and use it to collapse multi-experts into one merged FFN. **(Right):** Temporal feature modulation. We modulate the contribution of the temporal feature with the semantic association, which is measured by the similarity between the proxy text features retrieved from the fine-tuning and the test categories. The modulated embedding is used for inference.

## 3.2 Mixture-of-Temporal-Experts

**Architecture**    Consider the temporal module $h(\cdot|\theta_{tem})$ as $L$ repeated Transformer layers which consist of a self-attention layer and a fully-connected feed-forward network (FFN). For clarity, the temporal module's parameter $\theta_{tem}$ is factorized into parameter $\theta_{att}$ for attention layers and parameter $\theta_f$ for FFNs, where $\theta_{tem} = \theta_{att} \cup \theta_f$. Recent studies [5, 9] suggest that factual knowledge is mainly stored in the FFN, which consists of two learnable projection matrices and an activation function. Thus, we replace the FFN in each Transformer layer with $N$ temporal experts and obtain a set of experts $\{\{E_i^j\}_{i=1}^N\}_{j=1}^L$, where $E_i^j$ represents the $i^{th}$ expert at the $j^{th}$ layer and has the same structure as the FFN. Each expert starts training from different random initialization to ensure different optimization trajectories, which we experimentally show as critical for learning distinct knowledge (i.e. data bias views). Denote the parameter of the expert $E_i^j$ as $\theta_i^j$ and the activated expert's index at the $j^{th}$ layer as $\mathcal{E}(j)$. The training objective becomes:

$$\mathcal{L}_{\text{TE}} = \mathcal{L}(\theta_v, \theta_{att}, \boxed{\{\theta_{\mathcal{E}(j)}^j\}_{j=1}^L}; \mathcal{D}). \tag{4}$$

**Routing Policy**    The routing algorithm of MoE determines which experts process inputs. Classic routing mechanisms (i.e. top-$k$ gating [37]) make decisions using a learnable gate network and require auxiliary losses to handle load balancing. Instead, we adopt the stochastic routing algorithm [59] considering the architecture brevity, as it avoids the need for extra network, computation, and loss.

Recall that we expect each expert to learn distinct feature patterns. To further enlarge the discrepancy in knowledge learned by each expert, we present a stochastic routing algorithm based on the multinomial distribution. By assigning different activation probabilities to experts, we can control the training data volume of each expert, empowering their knowledge with different degrees of generalization and specialization. Formally, set $\mathbf{A} = [\alpha_i^l]_{i=1}^N$ to be a random vector in the $l^{th}$ layer, where $\alpha_i^l \in \{0, 1\}$ indicates whether the $i^{th}$ expert is activated and $\sum_{i=1}^N \alpha_i^l = 1$. We specify the variable $\mathbf{A}$ to follow the multinomial distribution where the activation probability of each expert is given as

$$P(\alpha_i^l = 1) = \frac{\exp(i)}{\sum_{i=1}^N \exp(i)}, \tag{5}$$

where $\exp(\cdot)$ is the natural exponential function. This allows experts with a greater index to be activated more likely and therefore receive a larger volume of data during training. Once the expert is selected, all inputs in a given batch are processed by the same expert.

**Knowledge Aggregation in Inference**  During inference, knowledge across experts can be aggregated by either merging weights [44, 47] or ensembling logits [16]. We adopt merging weights with the following benefits: (**i**) More effective knowledge aggregation as evidenced in Table 2. (**ii**) Constant computation cost, while the latter increases linearly with the expert number. (**iii**) Consistent parameter structure with its initial state. Specifically, for the experts $\{E_i^l\}_{i=1}^N$ in the $l^{th}$ layer, we average the weights of all the experts to obtain the final parameters $\tilde{\theta}^l$ for inference, where

$$\tilde{\theta}^l \leftarrow \frac{1}{N} \sum_{i=1}^N \theta_i^l. \tag{6}$$

## 3.3  Weight Merging Regularization

While the introduction of MoTE notably improves model generalization, it still incurs a decrease in close-set performance as presented in Table 1, which again proves the conflicting nature of generalization and specialization in weight space. We attribute this phenomenon to the difficulty of simultaneously preserving the generalized and specialized knowledge during expert merging process.

In this regard, we propose Weight Merging Regularization to reconcile and promote the aggregation efficacy of expert knowledge. To maximally aggregate the *specialized knowledge* of experts, the final merged model necessitates explicit optimization with respect to the task-specific objective. As for the preservation of the *generalization knowledge*, previous works [17, 22, 27] observe that the flatness (i.e. sharpness) of local minima in the loss landscape is strongly correlated with the model generalization. They suggest that models with flatter local minima in the loss landscape generalize better on unseen data, where the flatness refers to the robustness of the loss value to perturbations in the parameters. Within a flat basin of the loss landscape, moderate variations of parameters do not lead to significant increases in loss. Inspired by this observation, we propose to facilitate preserving the generalized knowledge of experts by explicitly constructing a *flat region* around the loss landscape of the final merged model. We optimize the constructed region with the task-specific loss objective and demonstrate that region construction benefits the coexistence of specialization and generalization in one unified model. Formally, for the experts $\{E_i^l\}_{i=1}^N$ in the $l^{th}$ layer, we first merge the weights as

$$\phi^l \leftarrow \sum_{i=1}^N \frac{\exp(i/\tau)}{\sum_{i'=1}^N \exp(i'/\tau)} \cdot \theta_i^l, \tag{7}$$

where $\tau$ is a temperature parameter used to control the softness of the distribution. By varying the value of $\tau$, we can build a region of the merged parameters in the weight space. Considering that sampling $\tau$ in a continuous space may lead to difficulties in model convergence, we sample $\tau$ from a discrete set given as $\{\pm 2^n \cdot \beta\}_{n=0}^4 \cup \{\infty\}$, where $\beta$ is a hyper-parameter of the candidate set. With parameters $\{\phi^j\}_{j=1}^L$ merged using different $\tau$ in each layer, the regularizer is defined as:

$$\mathcal{L}_{\text{WMR}} = \mathcal{L}(\theta_v, \theta_{att}, \boxed{\{\phi^j\}_{j=1}^L}; \mathcal{D}). \tag{8}$$

Denote the generated video feature when computing $\mathcal{L}_{\text{WMR}}$ as $\mathbf{z}_r$. We further slightly regularize the consistency between $\mathbf{z}_r$ and the pooled spatial feature $\mathbf{e}$ from the CLIP visual encoder:

$$\mathcal{L}_{\text{MSE}} = \mathbb{E}_{(V,C)\sim\mathcal{D}}[\|\mathbf{z}_r - \mathbf{e}\|_2^2], \tag{9}$$

where $\|\cdot\|_2$ is the L2 norm. This regularizer encourages MoTE to model the temporal dynamics with a light temporal feature, which potentially reduces the model complexity and improves generalization. The overall learning objective can be written as:

$$\mathcal{L}_{\text{ALL}} = \mathcal{L}_{\text{TE}} + \lambda\mathcal{L}_{\text{WMR}} + \eta\mathcal{L}_{\text{MSE}}, \tag{10}$$

where we set $\lambda = 0.5$ and $\eta = 0.1$ by default.

## 3.4  Temporal Feature Modulation

While applying MoTE to downstream recognition tasks, a potential risk is that its semantic space is limited to the category names of the fine-tuning dataset. To demonstrate this, we collect the category names of Kinetics-400, UCF-101, and Kinetics-600 datasets and manually remove duplicate ones. We

conduct tests with the mixed category names and observe 2.9% closed-set performance drops, 29.8% and 31.1% zero-shot performance drops on Kinetics-400, UCF-101, and Kinetics-600, respectively. That is, the model tends to classify all videos into fine-tuning categories.

In light of this, we propose a test-time adaptation strategy for networks where the temporal module is separated from the CLIP encoder, to measure the confidence of temporal features by means of CLIP's text space and modulates the contribution of temporal features to the prediction. Given the pooled feature $\mathbf{e}$ from the CLIP visual encoder, we generate fine-tuning and test proxy features $\mathbf{y}_f, \mathbf{y}_t \in \mathbb{R}^{K \times D}$ by retrieving the $K$ nearest text features in the fine-tuning and the test dataset categories. We estimate the semantic association $\rho$ between the two proxy features as

$$\rho = \exp(-(1 - \mathcal{M}(\mathbf{y}_t \mathbf{y}_f^T))/\gamma), \tag{11}$$

where $\gamma$ stands for a scale hyper-parameter and $\mathcal{M}(\cdot)$ is a sequential maximum and average pooling operation. Denote the pooled temporal feature from MoTE as $\mathbf{t}$, the final video embedding $\mathbf{z}$ used for inference becomes $\mathbf{z} = \mathbf{e} + \rho \cdot \mathbf{t}$.

# 4 Experiments

## 4.1 Experimental Setup

**Architecture.**   We employ the CLIP [33] pre-trained ViT-B/16 and ViT-L/14 in our experiments. On top of the visual encoder, we add 6 layers MoTE for ViT-L/14 and 4 layers MoTE for ViT-B/16, with 4 temporal experts per layer by default.

**Implementation Details.**   We fine-tune our model using the Kinetics-400 [15] dataset as in previous works [28]. During fine-tuning, we sparsely sample $T$ (e.g. 8 or 16) frames as the video input. Each input example is randomly cropped and resized to the size of $224 \times 224$ and then undergoes random horizontal flip and random grayscale. We adopt AdamW [25] as the optimizer with a weight decay of 0.2, following a half-period cosine learning rate decay. The initial learning rate is set to $5 \times 10^{-5}$ with a total batch size of 144. Furthermore, we set the candidate set $\beta$ for $\mathcal{L}_{\text{WMR}}$ to 0.6 and the scale parameter $\gamma$ for temporal feature modulation to 0.05, and $K$ to 5. We apply temporal feature modulation only in evaluation. *Please see supplementary for more details.*

**Evaluation Protocols.**   We thoroughly evaluate our method with close-set, zero-shot, and few-shot video recognition. *Close-set:* We evaluate the close-set performance on Kinetics-400 [15], using one single clip with a center crop (i.e. $1 \times 1$ views) or 4 clips with 3 crops (i.e. $4 \times 3$ views) per video [28]. Each view contains 8 or 16 sparsely sampled frames. *Zero-shot:* Following previous works [28, 34], we evaluate zero-shot performance on UCF-101 [38], HMDB-51 [19], and Kinetics-600 [3]. For K600, we adopt the three splits provided by [4]. Each split contains 160 categories out of 220 new categories. In zero-shot setting, we test using $3 \times 1$ views with 8 frames per view. *Few-shot:* We consider standard K-shot setting and evaluate on UCF-101, HMDB-51, and Something-Something v2 [10]. We adopt a single view for evaluation.

## 4.2 Ablation Studies

**Component-wise analysis of MoTE.**   In Table 1, we perform in-depth ablations of the proposed components with the ViT-L/14 network. We adopt Text4Vis [49] as our baseline, which serves as a prevalent CLIP adaptation framework in the video domain. Text4Vis uses a 6-layer Transformer for temporal modeling, featuring a high degree of specialization but low generalization capability. We observe that adopting the temporal experts boosts zero-shot performance significantly, validating our idea of improving generalization with multiple data bias views. We then introduce $\mathcal{L}_{\text{WMR}}$ for achieving the coexistence of generalization and specialization, which facilitates a more efficient aggregation of generalized knowledge while achieving the same level of specialization as Text4Vis. Adding $\mathcal{L}_{\text{MSE}}$ further improves the zero-shot performance. Moreover, we find that zero-shot performance benefits from modulating the temporal features for unseen categories during evaluation, especially for HMDB51. In summary, MoTE achieves a sound balance between generalization and specialization.

**Expert-wise performance of MoTE.**   To better understand the reconciliation effect of MoTE, we show the performance of each expert and the final merged model in Figure 3. In particular,

Table 1: Ablation study on various components of MoTE with ViT-L/14 network.

| Method | K400 | UCF | HMDB | K600 |
|---|---|---|---|---|
| Baseline (Text4Vis [49]) | 86.7 | 82.6 | 52.1 | 72.8 |
| +Temporal Experts | 85.5 | 86.6 | 56.3 | 78.3 |
| + $\mathcal{L}_{\text{WMR}}$ | **86.8** | 87.2 | 56.6 | 78.3 |
| + $\mathcal{L}_{\text{MSE}}$ | **86.8** | 87.5 | 57.0 | 78.9 |
| + Temp. Feature Modulation | **86.8** | **88.7** | **61.4** | **79.0** |

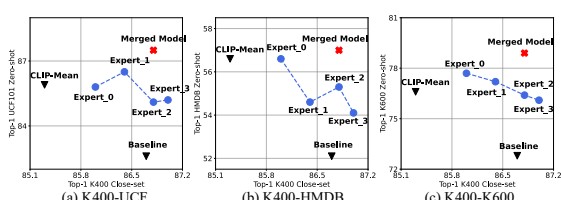

(a) K400-UCF  (b) K400-HMDB  (c) K400-K600

Figure 3: Expert-wise performance of MoTE. CLIP-Mean denotes a fine-tuned CLIP model with mean pooling for temporal modeling.

Table 2: Ablation studies on key details. We report close-set accuracy on K400 and zero-shot accuracy on UCF-101 and K600 split1, using the ViT-L/14 network. Default settings are colored in gray.

(a) Effect of initialization and training data across experts.

| Init. | Data | K400 | UCF | K600 |
|---|---|---|---|---|
| Same | All | 86.7 | 82.6 | 72.8 |
| Same | Routed | 86.3 | 86.4 | 77.9 |
| Different | All | **86.8** | 87.2 | **79.0** |
| Different | Routed | **86.8** | **87.5** | 78.9 |

(b) Varying numbers of temporal experts in each MoTE layer.

| Numbers | K400 | UCF | K600 |
|---|---|---|---|
| 3 | 86.6 | 86.7 | 77.4 |
| 4 | **86.8** | 87.5 | **78.9** |
| 5 | 86.7 | **87.6** | 78.5 |
| 6 | 86.7 | 87.1 | 78.3 |

(c) Various types of knowledge aggregation.

| Type | Param. | K400 | UCF | K600 |
|---|---|---|---|---|
| Random routing | 127.6M | 85.3 | 85.4 | 76.0 |
| Ensembling logits | 127.6M | 86.0 | 86.3 | 77.5 |
| Merging weights | 42.6M | **86.8** | **87.5** | **78.9** |

(d) Different routing policies during fine-tuning.

| Policies | K400 | UCF | K600 |
|---|---|---|---|
| Fixed | 86.7 | 82.6 | 72.8 |
| Random | **86.8** | 86.9 | 78.1 |
| Multinomial | **86.8** | **87.5** | **78.9** |

(e) Different types of $\mathcal{L}_{\text{WMR}}$ and candidate set parameter $\beta$.

| Type | $\beta$ | K400 | UCF | K600 |
|---|---|---|---|---|
| Point | - | 86.1 | 85.8 | 78.0 |
|  | 0.6 | **86.8** | **87.5** | **78.9** |
| Region | 0.8 | **86.8** | 87.2 | 78.7 |
|  | 1.0 | 86.7 | **87.5** | 78.5 |

(f) Varying scale parameters $\gamma$ for the temporal feature modulation.

| $\gamma$ | K400 | UCF | K600 |
|---|---|---|---|
| 0.04 | **86.8** | 88.6 | 78.9 |
| 0.05 | **86.8** | **88.7** | **79.0** |
| 0.06 | **86.8** | 88.6 | 78.8 |
| 0.07 | **86.8** | 88.2 | 78.6 |

CLIP-Mean denotes a fine-tuned CLIP model with mean pooling for temporal modeling. We do not apply temporal feature modulation in this study. As shown in the figure, each expert exhibits distinct zero-shot and close-set performances, suggesting that diverse knowledge is learned across experts, as expected in Section 3.2. We find that the zero-shot performance of each dataset presents different sweet spots in terms of the generalization degree. For example, Expert_1 achieves the best results on UCF-101, while Expert_0 performs best on HMDB-51. After weight merging using Equation 6, the merged model yields better zero-shot results than any expert and CLIP-Mean, and the close-set performance slightly outperforms Text4Vis. It demonstrates that our method effectively preserves the generalized and specialized knowledge of every expert, which can be attributed to $\mathcal{L}_{\text{WMR}}$.

**Effect of distinct optimization trajectories.** In Table 2a, we show that learning diverse knowledge from distinct optimization trajectories is critical for improving generalization. "Data" indicates whether the input data is sent to *all* experts or *routed* to one expert for encoding. In the first row, the optimization trajectories across experts are identical since all experts share the same initialization and input data. We find that applying different initialization and training data across experts can significantly improve generalization, respectively. When comparing row 3 with row 4, using the routing strategy achieves similar generalization performance while keeping the computation cost low.

**Varying numbers of experts and temporal layers** We ablate the number of experts in Table 2b. We observe that the zero-shot performance gains stabilize after a proper number of experts, and 4 temporal experts per layer are sufficient to capture various data bias views. As for the number of temporal layers, we obtain consistent gains with different numbers of layers as shown in Figure 1 (b), demonstrating the scalability of our method.

**Various types of knowledge aggregation.** As shown in Table 2c, the inferior performance of random routing indicates the necessity of leveraging diverse knowledge across experts. Compared with ensembling logits, merging weights aggregate knowledge more efficiently with fewer parameters. Meanwhile, the computation cost of ensembling logits is 4 × merging weights.

**Effect of different routing policies.** Table 2d ablates different routing policies. The fixed routing policy sends all inputs to a single expert. Our multinomial routing policy achieves the best performance since each expert learns more discrepant knowledge than random routing.

**Weight Merging Regularization.** We study different types of the merging regularizer $\mathcal{L}_{\text{WMR}}$ as well as candidate set parameters $\beta$ in Table 2e. "Point" indicates the regularizer with the average merged

Table 3: Zero-shot video recognition performance compared with the state-of-the-art methods on UCF-101, HMDB-51, and Kinetics-600. † denotes reproduced results with our implementation.

| Method | Venue | Encoder | Frames | UCF-101 | HMDB-51 | Kinetics-600 |
|---|---|---|---|---|---|---|
| ActionCLIP [42] | arXiv'21 | ViT-B/16 | 32 | 58.3±3.4 | 40.8±5.4 | 67.7±1.1 |
| A5 [14] | ECCV'22 | ViT-B/16 | 32 | 69.3±4.2 | 44.3±2.2 | - |
| X-CLIP [28] | ECCV'22 | ViT-B/16 | 32 | 72.0±2.3 | 44.6±5.2 | 65.2±0.4 |
| ST-Adapter† [30] | NeurIPS'22 | ViT-B/16 | 8 | 76.9±0.8 | 51.5±0.6 | 60.2±1.8 |
| Vita-CLIP [45] | CVPR'23 | ViT-B/16 | 8/32 | 75.0±0.6 | 48.6±0.6 | 67.4±0.5 |
| ViFi-CLIP [34] | CVPR'23 | ViT-B/16 | 32 | 76.8±0.7 | 51.3±0.6 | 71.2±1.0 |
| OTI [58] | ACMMM'23 | ViT-B/16 | 8 | 83.3±0.3 | 54.2±1.3 | 66.9±1.0 |
| Open-VCLIP [46] | ICML'23 | ViT-B/16 | 8 | **83.4**±1.2 | 53.9±1.2 | **73.0**±0.8 |
| MAXI [23] | ICCV'23 | ViT-B/16 | 16/32 | 78.2±0.8 | 52.3±0.7 | 71.5±0.8 |
| **MoTE (ours)** | | ViT-B/16 | 8 | **83.4**±0.7 | **55.8**±0.9 | 70.2±0.6 |
| X-Florence [28] | ECCV'22 | Florence | 32 | 73.2±4.2 | 48.4±4.9 | 68.8±0.9 |
| Text4Vis† [49] | AAAI'23 | ViT-L/14 | 8 | 82.6±0.7 | 52.4±0.4 | 72.1±0.9 |
| OTI [58] | ACMMM'23 | ViT-L/14 | 8 | 88.1±1.0 | 59.3±1.7 | 70.6±0.5 |
| Open-VCLIP [46] | ICML'23 | ViT-L/14 | 8 | 87.6±1.2 | 59.0±0.6 | **81.1**±0.8 |
| DiST† [32] | ICCV'23 | ViT-L/14 | 32 | 74.9±0.8 | 57.5±1.6 | 75.0±0.7 |
| **MoTE (ours)** | | ViT-L/14 | 8 | **88.7**±0.6 | **61.4**±1.3 | 78.4±0.9 |

parameters (i.e. without region construction). Due to the conflicting nature of generalization and specialization in optimization, the regularizer without region construction cannot adequately aggregate expert knowledge. Instead, constructing the region in the loss landscape enables generalization and specialization to coexist in one model. The parameter $\beta$ is used to construct the candidate set of the weights merging temperature. As shown in the table, the performance gain is robust to this parameter.

**Varying numbers of $\gamma$ for Temporal Feature Modulation.** The parameter $\gamma$ is used to scale the semantic association to an appropriate interval. In general, $\gamma = 0.05$ yields better results. We find that different datasets exhibit varying sensitivities to this parameter and a relatively small contribution of temporal features generalizes to the unseen categories. *Please see supplementary for more ablations.*

### 4.3 Main Results

**Zero-shot video recognition.** In Table 3, we compare our method with the state-of-the-art results under the zero-shot setting. Our method achieves new state-of-the-art results on UCF-101 and HMDB-51, with only 8 frames as input. The remarkable performance can be scaled up as the network size and the input frame number increase, demonstrating the great scalability of our method. For K600, although significant improvements are achieved over the baseline, our method still underperforms some methods that do not apply any additional parameters such as Open-VCLIP [46] and MAXI [23]. Since the categories of Kinetics-600 are much more complex, we argue that it is still challenging for randomly initialized parameters to generalize well on such complex unseen categories. Note that our method still outperforms any method that employs additional parameters on K600. Overall, our method presents superior generalization performance.

**Close-set and zero-shot performance trade-off.** Table 4 presents comparisons with the state-of-the-art methods under the close-set setting. We also list the harmonic mean of the zero-shot results on UCF, HMDB, and K600 as an indicator of the generalization capability, denoted as $HM_{ZS}$. To evaluate the holistic performance in close-set and zero-shot settings, we define the "Trade-off" score as the harmonic mean of Top-$1_{K400}$ and $HM_{ZS}$, as we consider the specialization and generalization capabilities to be equally important for a balanced model.

As shown in the table, our method presents strong close-set results competitive with SOTAs and state-of-the-art zero-shot performance. More importantly, while existing methods can perform well on only one task setting, we achieve superior performance on both settings simultaneously with one unified model. Our method consistently exhibits significant trade-off performance advantages across different networks, evidencing the effectiveness and scalability of MoTE in reconciling generalization and specialization capabilities. Even against ViFi-CLIP [34] which uses different training hyperparameters for close-set and zero-shot settings and takes more frames as the input, our method still shows better balanced performance (74.7% vs. 72.9%). Noteworthy that FROSTER [12] is a strong zero-shot action recognition model but performs less well in the close-set setting, which

Table 4: Close-set and zero-shot performance trade-off compared with the state-of-the-art methods. We report close-set results on K400. "HM$_{ZS}$" indicates the harmonic mean of zero-shot results on UCF, HMDB, and K600. "Trade-off" is defined as the harmonic mean of Top-1$_{K400}$ and HM$_{ZS}$. "Unified model" indicates whether the method is evaluated using the same model in both settings.

| Method | Encoder | Input Size | GFLOPs × Views | Param (M) | Unified model | K 400 Top-1 | K 400 Top-5 | HM$_{ZS}$ | Trade-off |
|---|---|---|---|---|---|---|---|---|---|
| A6 [14] | ViT-B/16 | 16×224 | - | 91.2 | - | 76.9 | 93.5 | - | - |
| X-CLIP [28] | ViT-B/16 | 8×224 | 145×1×1 | 131.5 | ✗ | **82.3** | 95.8 | 58.1 | 68.1 |
| Vita-CLIP [45] | ViT-B/16 | 8×224 | /×1×1 | 125.0 | ✓ | 80.5 | 95.9 | 61.5 | 69.7 |
| Open-VCLIP [46] | ViT-B/16 | 8×224 | /×1×1 | 86.2 | ✓ | 78.9 | - | 67.8 | 72.9 |
| FROSTER [12] | ViT-B/16 | 8×224 | /×1×1 | 86.2 | ✓ | 78.9 | 94.8 | **69.1** | 73.7 |
| **MoTE (ours)** | ViT-B/16 | 8×224 | 141×1×1 | 98.8 | ✓ | 81.8 | **95.9** | 67.9 | **74.2** |
| ActionCLIP [42] | ViT-B/16 | 16×224 | 282×10×3 | 141.7 | ✗ | 82.6 | 96.2 | 53.2 | 64.7 |
| SiF [43] | ViT-B/16 | 8×224 | 142×4×3 | 143.9 | - | 77.4 | 93.6 | - | - |
| X-CLIP [28] | ViT-B/16 | 8×224 | 145×4×3 | 131.5 | ✗ | 83.8 | **96.7** | 58.1 | 68.6 |
| ST-Adapter [30] | ViT-B/16 | 8×224 | 152×3×1 | 93.0 | ✓ | 82.0 | 95.7 | 61.2 | 70.1 |
| Text4Vis [49] | ViT-B/16 | 8×224 | 141×4×3 | 105.2 | ✓ | 82.9 | - | 59.1 | 69.0 |
| Vita-CLIP [45] | ViT-B/16 | 8×224 | /×4×3 | 125.0 | ✓ | 81.8 | 95.9 | 61.5 | 70.2 |
| ViFi-CLIP [34] | ViT-B/16 | 16×224 | 281×4×3 | 149.6 | ✗ | **83.9** | 96.3 | 64.4 | 72.9 |
| **MoTE (ours)** | ViT-B/16 | 8×224 | 141×4×3 | 98.8 | ✓ | 83.0 | 96.3 | 67.9 | **74.7** |
| X-Florence [28] | Florence | 32×224 | 2822×4×3 | - | ✗ | 86.5 | 96.9 | 61.4 | 71.8 |
| AIM [53] | ViT-L/14 | 8×224 | 934×3×1 | 341.0 | - | 86.8 | 97.2 | - | - |
| Text4Vis [49] | ViT-L/14 | 8×224 | 649×4×3 | 346.6 | ✓ | 86.7 | 97.4 | 66.6 | 75.3 |
| Open-VCLIP [46] | ViT-L/14 | 8×224 | /×3×1 | 304.0 | ✓ | 83.9 | 96.5 | 73.7 | 78.5 |
| DiST [32] | ViT-L/14 | 8×224 | 710×3×1 | 343.0 | ✓ | **86.9** | **97.6** | 68.1 | 76.4 |
| **MoTE (ours)** | ViT-L/14 | 8×224 | 649×4×3 | 346.6 | ✓ | 86.8 | 97.5 | 74.4 | **80.1** |
| **MoTE (ours)** | ViT-L/14 | 16×224 | 1299×4×3 | 346.6 | ✓ | 87.2 | 97.7 | 74.8 | 80.5 |

Table 5: Few-shot results compared with the state-of-the-art methods on HMDB, UCF, and SSv2. All methods directly fine-tune on CLIP, except MAXI (Tuning after pre-training on K400.).

| Method | HMDB-51 $K$=2 | $K$=4 | $K$=8 | $K$=16 | UCF-101 $K$=2 | $K$=4 | $K$=8 | $K$=16 | SSv2 $K$=2 | $K$=4 | $K$=8 | $K$=16 |
|---|---|---|---|---|---|---|---|---|---|---|---|---|
| CLIP [33] | 47.2 | 47.2 | 47.2 | 47.2 | 75.1 | 75.1 | 75.1 | 75.1 | 2.9 | 2.9 | 2.9 | 2.9 |
| A6 [14] | 39.7 | 50.7 | 56.0 | 62.4 | 71.4 | 79.9 | 85.7 | 89.9 | 4.4 | 5.1 | 6.1 | 9.7 |
| X-CLIP [28] | 53.0 | 57.3 | 62.8 | 64.0 | 76.4 | 83.4 | 88.3 | 91.4 | 3.9 | 4.5 | 6.8 | 10.0 |
| ActionCLIP [42] | 47.5 | 57.9 | 57.3 | 59.1 | 70.6 | 71.5 | 73.0 | 91.4 | 4.1 | 5.8 | 8.4 | 11.1 |
| ViFi-CLIP [34] | 57.2 | 62.7 | 64.5 | 66.8 | 80.7 | 85.1 | 90.0 | 92.7 | 6.2 | 7.4 | 8.5 | **12.4** |
| MAXI [23] | 58.0 | 60.1 | 65.0 | 66.5 | 86.8 | 89.3 | **92.4** | 93.5 | 7.1 | 8.4 | 9.3 | **12.4** |
| **MoTE (ours)** | **61.3** | **63.9** | **67.2** | **68.2** | **88.8** | **91.0** | 92.3 | **93.6** | **7.3** | **8.5** | **9.5** | 12.2 |
| | +3.3 | +1.2 | +2.2 | +1.4 | +2.0 | +1.7 | - | +0.1 | +0.2 | +0.1 | +0.2 | - |

can be attributed to the the application of the weights ensemble and the distillation supervision from the Frozen CLIP limits the model's ability to learn video-specialized knowledge.

**Few-shot video recognition.** We perform all-way few-shot video recognition in Table 5, which requires both specialization and generalization to rapidly adapt to a novel set of categories with limited samples. Our method presents consistent improvements across sample shots in all datasets, demonstrating the strong learning capacity and transferability. Our method even outperforms MAXI [23] which uses K400 fine-tuning weights as initialization and more diverse textual knowledge.

### 4.4 Category-wise performance visualization.

To better understand the specific video categories that each expert excels at recognizing, we visualize the Top-1 classification accuracy for categories sampled from the UCF-101 dataset in Figure 4. The experiment is conducted in the zero-shot setting. We observe that different experts show distinct performance across video categories, likely due to the diverse generalization knowledge learned by each expert. The merged expert benefits from the aggregation of expert knowledge, particularly for categories where temporal information is crucial, such as 'Handstand Pushups' and 'Wall Pushups'. This suggests that each expert can capture various temporal patterns within the same category.

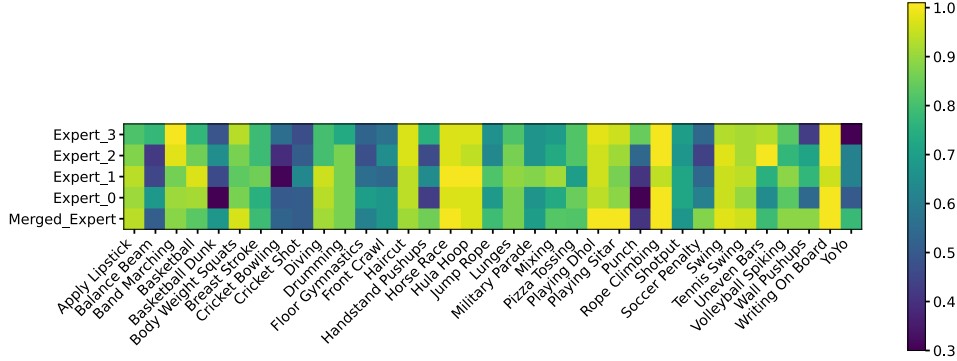

Figure 4: Visualization of the Top-1 accuracy for each video category sampled from UCF-101 with respect to the merged expert and each individual expert.

## 5 Related Work

**Video recognition.** In the era of deep learning, early works explored diverse variants of convolutional networks for joint spatiotemporal modeling, such as 3D convolution [2, 39] and factorized spatial and temporal convolution [40]. The advent of Transformer then attracted the attention of researchers and has been widely applied for video recognition [1, 6]. In addition, self-supervised learning methods learn transferable representations from large-scale unlabeled data and achieve promising performances [7, 31, 57].

**Transferring VLMs for videos.** Transferring VLMs for video recognition task has been proven to be effective. ViFi-CLIP [34] suggests that a direct fine-tuning process generalizes well on various settings. Open-VCLIP [46] constructs an open-vocabulary video model by interpolating the model weights and its optimization trajectory. Vita-CLIP [45] extracts discriminative information using multi-level prompts. X-CLIP [28] proposes cross-frame attention and multi-frame integration modules for temporal modeling. Text4Vis [49] and BIKE [50] explore the way for more effective knowledge transfer and use multi-transformer layers to capture temporal cues. FROSTER [12] mitigates catastrophic forgetting by ensuring the learned features do not diverge too far from the frozen ones through distillation. This divergence comes from the variations of both CLIP and additional trainable parameters. Differently, we believe the main cause of the forgetting problem is the overfitting of additional parameters regardless of whether the CLIP parameters are tuned. Note that FROSTER can also potentially improve the generalization of additional parameters through knowledge distillation, but our method presents a more explicit way to achieve this goal. Existing methods face a trade-off of introducing more specialized knowledge or preserving more generalized knowledge. Our work addresses this challenge and presents superior results on both close-set and zero-shot settings.

**Mixture-of-Experts.** The sparsely activated MoE structure [37] enables the model capacity (i.e. number of parameters) to be vastly scaled up while keeping the computation cost per sample basically unchanged. This technique has been widely investigated in building large-scale pre-trained models in various fields [20, 26, 35]. Several works employ MoE on large-scale language models for parameter-efficient tuning to improve their specialized performance [8, 44]. In this paper, we demonstrate its effectiveness in balancing generalization and specialization in VLM knowledge transfer.

## 6 Conclusion

In this paper, we present MoTE, an effective Visual-Language to video knowledge transfer framework that enjoys both superior generalization and specialization. MoTE leverages a mixture of temporal experts to enhance performance in both close-set and zero-shot video recognition, all while maintaining the conventional temporal module's structure and computational efficiency. With the proposed weight merging regularization and temporal feature modulation, we achieve the coexistence of generalization and specialization in one unified model. Extensive experiments validate MoTE's ability to strike an optimal trade-off between close-set and zero-shot performance.

## Acknowledgements

This paper is supported by the National Natural Science Foundation of China under Grants (62073245, 62173248, 62233013), Shanghai Municipal Science and Technology Major Project (2021SHZDZX0100) and Innovation Action Plan (22511104900), the Fundamental Research Funds for the Central Universities.

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

## Supplemental material

## A Limitation and Broader Impact

**Limitation and future work**   While our method yields superior results in various settings of video recognition, there are still several aspects for improvement. First, the text space of MoTE is limited to video category names, which do not provide discriminative and semantically rich textual representations. Thus, exploring how to extend the semantic space with large-scale generative models may further enhance the performance of MoTE. Besides, additional parameters take various forms when adapting VLM to downstream tasks. In future work, we will further extend our approach to other forms of parameters to explore the generality and versatility of our method.

**Broader Impact**   Adapting foundation models to downstream tasks has become a dominant paradigm in machine learning [24, 30], we believe that it is worthwhile and important to explore how to preserve the existing knowledge of the foundation model when introducing new knowledge. We hope this work brings new insights for the broader and long-term adaptation of foundation models. Besides, our work focuses on video recognition tasks, which have a wide range of real-world application scenarios, such as surveillance. However, this work requires careful elimination of privacy and rights concerns before real-world deployments.

Table 6: Hyper-parameter details during fine-tuning.

|  | Value |
|---|---|
| *Optimization details* |  |
| Batch size | 144 |
| Optimizer | AdamW |
| Weight decay | 0.2 |
| Adam $\beta_1,\beta_2$ | 0.9, 0.999 |
| Learning rate (Base) | 5e-5 |
| Learning rate (CLIP layers) | 3e-6 |
| Learning rate decay | Cosine schedule |
| Training epochs | 30 (ViT-B), 20 (ViT-L) |
| Linear warm-up epochs | 5 |
| *Augmentation* |  |
| RandomResizedCrop |  |
|     Area | [0.08, 1.00] |
|     Aspect ratio | [3/4 , 4/3] |
|     Crop size | 224 |
| Random Horizontal Flip | 0.5 |
| Random Gray scale | 0.2 |

## B More Implementation Details

**Close-set and zero-shot video recognition.**   In Table 6, we present the hyper-parameters set for optimization. Note that we train one model for both close-set and zero-shot tasks, rather than training for each task separately. The parameters of all projection matrices in MoTE are initialized with values drawn from the normal distribution (mean=0, std=0.02). All bias terms are initialized as zero. We conduct experiments with 3 NVIDIA GeForce RTX 4090.

For zero-shot evaluation, the methods are evaluated on three official splits of UCF-101 and HMDB-51. For Kinetics-600, we adopt the three splits provided by [4]. Each split contains 160 categories out of 220 new categories that do not exist in K400. We report the average Top-1 accuracy and the standard deviation on three splits.

**Few-shot video recognition.**   We consider the standard K-shot setting [28, 34], where 2, 4, 8, and 16 video data are randomly sampled for each category for constructing the training dataset. Following previous work [49], we repeat the constructed training dataset to maintain the same size as the

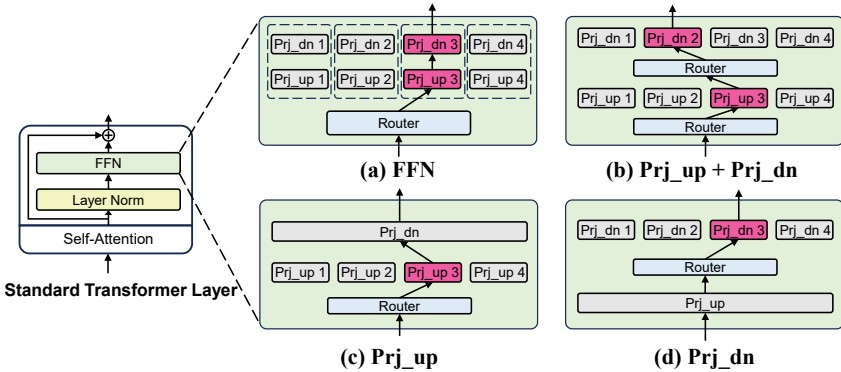

Figure 5: Illustration of optional architecture designs for the temporal expert. We omit the activation function between the projection matrices for brevity.

original full dataset. In few-shot settings, we fine-tune models for 2 epochs on SSv2 and 10 epochs on UCF-101 and HMDB-51. We adopt a single view for evaluation.

Table 7: Additional ablation studies. Default settings are colored in gray.

(a) Optional designs for the temporal expert.

| Experts | K400 | UCF | K600 |
|---|---|---|---|
| Prj_up+Prj_dn | 86.7 | 84.2 | 74.1 |
| Prj_up | 86.1 | 83.3 | 73.6 |
| Prj_dn | 86.3 | 83.1 | 73.2 |
| FFN | **86.8** | **87.5** | **78.9** |

(b) Varying neighbor numbers $K$ for the temporal feature modulation.

| $K$ | K400 | UCF | K600 |
|---|---|---|---|
| 1 | **86.8** | **88.9** | 78.9 |
| 5 | **86.8** | 88.7 | **79.0** |
| 10 | **86.8** | 88.5 | 78.7 |

(c) Different types of Weight Merging Regularization.

| Types | K400 | UCF | K600 |
|---|---|---|---|
| $\mathcal{L}_{\text{WMR}}$ | **86.8** | **87.5** | **78.9** |
| $\mathcal{L}_{\text{WMR\_KL}}$ | 86.0 | 83.1 | 72.5 |
| $\mathcal{L}_{\text{WMR\_MSE}}$ | 86.3 | 86.2 | 76.3 |

(d) Ablation study on the training costs of MoTE.

| Method | GPU-days | GFLOPs |
|---|---|---|
| Baseline | 4.35 | 649 |
| +Temporal Experts | 4.35 | +0.34 |
| + $\mathcal{L}_{\text{WMR}}$ | 4.50 | +0.34 |
| + $\mathcal{L}_{\text{MSE}}$ | 4.51 | - |

(e) Results of fine-tuning on UCF and zero-shot evaluating on K400.

| Method | UCF | K400 |
|---|---|---|
| CLIP | 80.9 | 59.2 |
| Baseline | 95.6 | 56.6 |
| MoTE | **96.1** | **66.3** |

(f) Effect of temperature selection schemes.

| Type | K400 | K600 |
|---|---|---|
| Discrete set | **86.8** | **78.9** |
| Continuous normal dist. | 86.5 | 77.4 |
| Continuous uniform dist. | 86.0 | 77.9 |

## C Additional Ablations

**Optional architecture of temporal expert.** By default, we replace the whole FFN with experts of the same structure as the FFN. Optionally, we can replace one or both of the projection matrices in the FFN with experts of the same structure, as illustrated in Figure 5. We study the optional architecture design of the temporal expert in Table 7a. As shown in the table, we observe degraded performance with projection-level expert designs. We argue that the shared projection matrix or two-stage routing design of the projection-level expert allows knowledge communication between experts, such implicit communication may lead to mutual fitting between the experts. Therefore, preventing the knowledge exchange across experts is critical for learning diverse generalized knowledge.

**Varying neighbor numbers $K$ for Temporal Feature Modulation.** In Section 3.4 of the main manuscript, we generate fine-tuning proxy features $\mathbf{y}_f$ and test proxy features $\mathbf{y}_t$ by retrieving the $K$ nearest text features in the fine-tuning and the test dataset categories. We study the influence of various $K$ in Table 7b. We find that a large $K$ leads to a slight performance degradation as the retrieved text features may not represent the semantic information of the video properly. Overall, our performance gains are stable for this parameter.

**Various Loss Types of Weight Merging Regularization.** In Section 3.3 of the main manuscript, we design the weight merging regularization $\mathcal{L}_{\text{WMR}}$ based on the cross-entropy loss function. In this study, we use other loss function types to redesign the weight merging regularization $\mathcal{L}_{\text{WMR}}$. Note that our training procedure requires two forward passes of the temporal module, one when

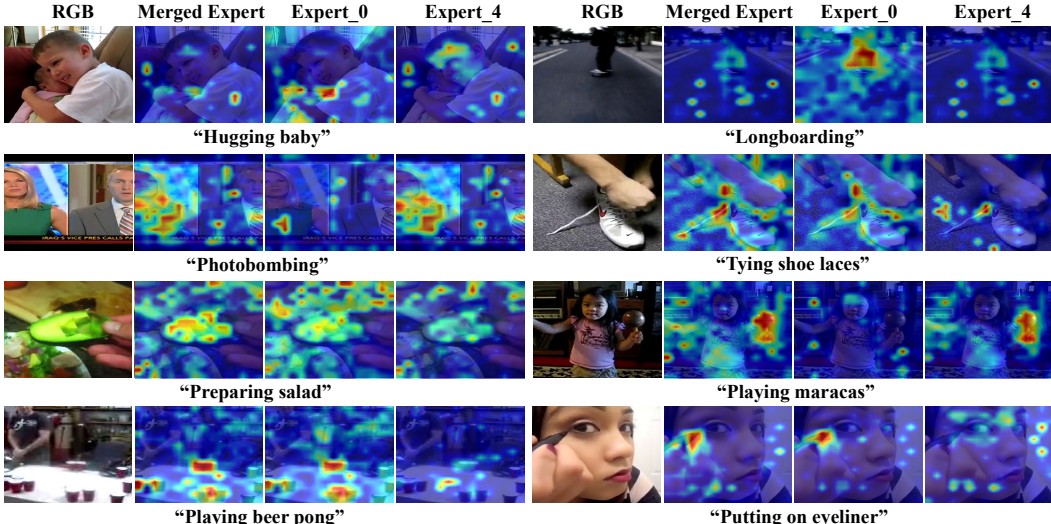

| RGB | Merged Expert | Expert_0 | Expert_4 | RGB | Merged Expert | Expert_0 | Expert_4 |

"Hugging baby"      "Longboarding"

"Photobombing"      "Tying shoe laces"

"Preparing salad"      "Playing maracas"

"Playing beer pong"      "Putting on eyeliner"

Figure 6: Visualization of attention maps. We show the RGB image and the attention maps of the merged expert, expert_0, and expert_4.

optimizing the activated expert with $\mathcal{L}_{\text{TE}}$ and the second in calculating weight merging regularization $\mathcal{L}_{\text{WMR}}$. When we optimize $\mathcal{L}_{\text{WMR}}$ using cross entropy, we use the ground truth label of the video as the supervised signal. We try to redesign $\mathcal{L}_{\text{WMR}}$ with other supervisions. (i) KL Divergence: We adopt the logits generated during the computation of $\mathcal{L}_{\text{TE}}$ as the supervision and optimize with KL divergence loss, denoted as $\mathcal{L}_{\text{WMR\_KL}}$. (ii) Mean Square Error: We use the output feature when computing $\mathcal{L}_{\text{TE}}$ as the supervision and optimizing with MSE loss, denoted as $\mathcal{L}_{\text{WMR\_MSE}}$. The results are presented in Table 7c. Interestingly, we find that the MSE loss achieves notable results in a weakly supervised manner, which indicates the potential scalability of our proposed approach.

**Training cost analysis of MoTE.** We report the actual training time of our method with respect to the baseline in Table 7d. The wall-clock time of training is benchmarked on 3 4090 GPUs with a batch size of 144. GPU days are calculated by the number of GPUs multiplied by the training time in days. As shown in the table, applying the mixture of temporal experts does not introduce additional training overhead over baseline, which can be attributed to the use of the routing strategy. Adding $\mathcal{L}_{\text{WMR}}$ and $\mathcal{L}_{\text{MSE}}$ brings a +0.16 days training time increase since it requires an additional forward pass of the temporal module.

**Transferring CLIP with small-scale fine-tuning data.** In the main text, our experimental setting follows previous works in fine-tuning on large-scale K400 and then evaluating zero-shot performance on relatively small downstream datasets. To further investigate MoTE's capabilities when the fine-tuning data are limited, we train the ViT-L/14 network on UCF-101 and evaluate it on K400, results are shown in Table 7e. MoTE yields a 0.5 improvement over the Baseline on UCF-101, while zero-shot performance on K400 significantly outperforms both raw CLIP and Baseline. This demonstrates the applicability of MoTE to small-scale datasets and its ability to learn generalized knowledge from limited data.

**Effect of temperature selection schemes.** In this study, we compare our discrete temperature sampling strategy with the continuous space selection schemes. We implement two continuous space selection schemes. (1) Sampling from a continuous standard normal distribution (mean=0, variance=1). (2) Sampling from a continuous uniform distribution. As shown in Table 7f, the continuous space sampling strategy results in a notable performance degradation.

# D    Textual Prompt Templates

In our work, we adopt a set of hand-craft textual prompt templates to generate text embeddings. Following CLIP [33], we perform prompt ensembling over the 28 templates in order to provide comprehensive semantics. The templates are listed in Table 8.

Table 8: Textual prompt templates of MoTE.

| Templates |
| --- |
| 'a photo of {category}.' |
| 'a photo of a person {category}.' |
| 'a photo of a person using {category}.' |
| 'a photo of a person doing {category}.' |
| 'a photo of a person during {category}.' |
| 'a photo of a person performing {category}.' |
| 'a photo of a person practicing {category}.' |
| 'a video of {category}.' |
| 'a video of a person {category}.' |
| 'a video of a person using {category}.' |
| 'a video of a person doing {category}.' |
| 'a video of a person during {category}.' |
| 'a video of a person performing {category}.' |
| 'a video of a person practicing {category}.' |
| 'a example of {category}.' |
| 'a example of a person {category}.' |
| 'a example of a person using {category}.' |
| 'a example of a person doing {category}.' |
| 'a example of a person during {category}.' |
| 'a example of a person performing {category}.' |
| 'a example of a person practicing {category}.' |
| 'a demonstration of {category}.' |
| 'a demonstration of a person {category}.' |
| 'a demonstration of a person using {category}.' |
| 'a demonstration of a person doing {category}.' |
| 'a demonstration of a person during {category}.' |
| 'a demonstration of a person performing {category}.' |
| 'a demonstration of a person practicing {category}.' |

# E    Qualitative Analysis

## E.1    Model Attention

To better understand what knowledge the experts capture, we present the visualization of the model attention in Figure 6. We show the attention map of the merged expert, expert_0, and expert_4 to explore whether diverse knowledge is learned across experts. Since MoTE only requires the input of frame-level tokens from the CLIP visual encoder, we calculate the attention map between the temporal video features output by MoTE and the image patch tokens from the CLIP encoder. As shown in the figure, experts_0 and experts_4 always focus on different regions, indicating that they capture various temporal patterns. Besides, we observe that the merged expert is able to focus on more precise and broader foreground areas. This phenomenon suggests that the merged expert sufficiently aggregates and leverages the knowledge from different experts.

## E.2    Representation similarity across experts.

We visualize the representation similarities across each expert and the final merged model in Figure 7. The representation similarities are averaged on 100 randomly sampled data of unseen categories from the K600 dataset. As show in the affinity map, the different similarities demonstrate that each expert learns distinctive knowledge from different optimization trajectories. Besides, we observe that the similarities between the merged model and each expert are relatively stable, indicating that the merged expert efficiently leverages the knowledge contained in each expert.

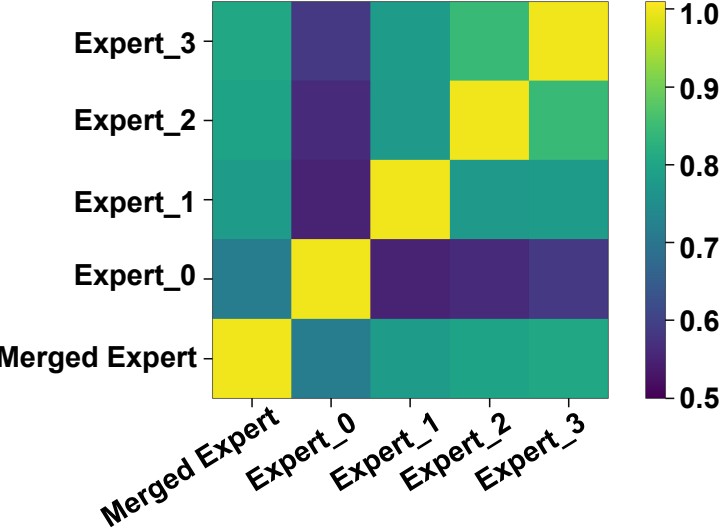

Figure 7: Visualization of representation similarities across each expert and the final merged model.

## F  Dataset Details

**Kinetics-400 [15]**   is a large-scale dataset in the video domain. The dataset contains ∼240k training videos and ∼20k validation videos in 400 human action categories, with an average length of 10 seconds. The high quality of the dataset makes it the most popular benchmark for video recognition

**Kinetics-600 [3]**   is an extension of Kinetics-400, consisting of ∼392k training videos, ∼30k validation videos, and ∼60k test videos in 600 human action categories. The dataset contains an additional 220 new action categories over Kinetics-400. We evaluate the zero-shot performance on 220 new categories and adopt three splits provided by the previous work [4]. We use its test set for evaluation and report the average performance on three splits.

**UCF-101 [38]**   is an action recognition dataset that contains 13,320 videos in 101 action categories, collected from YouTube. There are three official splits of training data and validation data.

**HMDB-51 [19]**   contains 7k videos in 51 action categories, collected from movie clips and web videos. There are three official splits of the dataset, each with 3,570 training data and 1,530 validation data. is a collection of realistic videos from various sources, including movies and web videos. The dataset comprises 7,000 video clips from 51 action categories.

**Somethin-Something V2 [10]**   is a temporal-heavy dataset that requires the fine-grained temporal understanding capability of the model. It contains 220,000 videos in 174 action categories.

