# OpenReview forum: "MoTE: Reconciling Generalization with Specialization for Visual-Language to Video Knowledge Transfer"
_NeurIPS.cc/2024/Conference — NeurIPS 2024 poster_

### Official Review · Reviewer_XfMx · 2024-07-10

**Soundness:** 2
**Presentation:** 3
**Contribution:** 2
**Rating:** 6
**Confidence:** 3

**Summary:**

The paper introduces a novel framework called MoTE. This framework addresses the trade-off between zero-shot generalization and close-set performance in video recognition tasks by tuning a mixture of temporal experts. The key contributions include:

- Introducing Weight Merging Regularization to balance generalization and specialization.
- Proposing temporal feature modulation to improve generalization during inference.
- Demonstrating state-of-the-art or competitive results on various video datasets such as Kinetics-400, Kinetics-600, UCF-101, and HMDB-51.

**Strengths:**

- The introduction of Weight Merging Regularization and temporal feature modulation provides a novel approach to balancing generalization and specialization in video recognition.
- The experimental results are thorough, demonstrating the effectiveness of the proposed methods on multiple datasets.

**Weaknesses:**

- The framework's text space is confined to video category names, which limits the richness of textual representations. Expanding the semantic space using large-scale generative models could enhance performance.
- The method currently explores limited forms of additional parameters. Extending the approach to other forms could improve generality and versatility.
- While results on certain benchmarks are promising, the model's performance on more diverse and challenging datasets needs further validation.
- The additional complexity from Weight Merging Regularization and other components can slightly increase training time, which may be a barrier for real-time applications.
- Extensive fine-tuning required for different tasks can be computationally expensive and time-consuming.

**Questions:**

- Can you provide more details on how expanding the text space with large-scale generative models might improve the model's performance?
- How does the performance of MoTE vary with different numbers of temporal layers and experts? Are there optimal configurations for specific tasks?
- What measures can be taken to reduce the computational overhead introduced by the additional components such as Weight Merging Regularization?

**Limitations:**

The following work is recommended for citation & discussion:

Oh, C., Lim, H., Kim, M., Han, D., Yun, S., Choo, J., Hauptmann, A., Cheng, Z.-Q., & Song, K. (2023). Towards calibrated robust fine-tuning of vision-language models. In NeurIPS 2023 Workshop on Distribution Shifts: New Frontiers with Foundation Models.

Tu, S., Dai, Q., Wu, Z., Cheng, Z.-Q., Hu, H., & Jiang, Y.-G. (2023). Implicit temporal modeling with learnable alignment for video recognition. In Proceedings of the IEEE/CVF International Conference on Computer Vision (pp. 19936-19947).

---

> ### Author Rebuttal · Authors · 2024-08-06
>
> We appreciate the valuable comments and the time you have dedicated to this paper. Please find the responses to your comments below.
> ***
> **1. Expanding the semantic space with large-scale generative models.**
>
> Thanks for your constructive suggestion! Following your suggestion, we replace the category names with rephrased detailed descriptions, which are generated by a large generative language model GPT3.5 as used in FROSTER [1]. The generated descriptions expand the semantic space by providing more details about the action and the scene. For example, "*Ice climbing*" is rephrased to "*A person is seen climbing up a wall of ice using specialized equipment like crampons and ice axes*". The results are presented in the table below.
>
> |Method|K400 (close-set)|UCF|K600|
> |-|:-:|:-:|:-:|
> |MoTE + category name|86.8|88.7|79.0|
> |MoTE + rephrased description|86.8|89.2|80.2|
>
> We can see that the enriched description leads to better generalization performance. We will actively explore this direction in future work.
>
> [1] FROSTER: Frozen CLIP Is A Strong Teacher for Open-Vocabulary Action Recognition. ICLR 2024.
>
> ***
> **2. Extending the approach to other forms could improve generality and versatility.**
>
> Thanks for your constructive suggestion! To further demonstrate the generality and versatility of MoTE, we apply our method to ST-Adapter [2]. For simplicity, we remove the depth-wise Conv3D operation in our adapter. In this case, our baseline is slightly lower than ST-Adapter, but it doesn't affect justifying the effectiveness of our method. The results are shown in the table below.
>
> |Method|K400 (close-set)|UCF|K600|
> |-|:-:|:-:|:-:|
> |ST-Adapter|81.1|77.3|61.1|
> |ST-Adapter+MoTE|81.2|79.4|66.7|
>
> Our method delivers a notable generalization performance improvement while maintaining the close-set result, indicating the effectiveness of MoTE applied to alternative networks.
>
> [2] ST-Adapter: Parameter-Efficient Image-to-Video Transfer Learning. NeurIPS 2022.
>
> ***
> **3. The model's performance on more diverse and challenging datasets needs further validation.**
>
> Thanks for your suggestion! We have evaluated our method on the Something-Something V2 (SSv2) dataset under the few-shot setting in Table 5, which requires intensive temporal understanding capability of the model. We find an inappropriate hyperparameter setting during fine-tuning and present the fixed results in the table below.
>
> |Method|SSv2-2 shot|SSv2-4 shot|SSv2-8 shot|SSv2-16 shot|
> |-|:-:|:-:|:-:|:-:|
> |ViFi-CLIP|6.2|7.4|8.5|12.4|
> |MAXI|7.1|8.4|9.3|12.2|
> |MoTE|7.4|9.5|13.3|16.7|
>
> We also evaluate our method on SSv2 under the zero-shot setting, as shown in the table below
>
> |Method|SSv2-zero shot|
> |-|:-:|
> |CLIP|2.7|
> |ViFi-CLIP|4.5|
> |MoTE|6.4|
>
> In both settings, our method outperforms the previous works by a notable margin, demonstrating the effectiveness of our method in facing more challenging datasets.
>
> ***
> **4. The additional complexity from Weight Merging Regularization and other components can slightly increase training time.**
>
> Thanks for your thoughtful comment! We show the training time costs (Table 8 of the supplementary material) and the corresponding GFLOPs in the table below. GPU days are calculated by the number of GPUs multiplied by the training time in days.
>
> |Method|GPU-days|GFLOPs|
> |-|:-:|:-:|
> |Baseline|4.35|649|
> |+MoTE|4.35|+0.34|
> |+$L_{WMR}$|4.50|+0.34|
> |+$L_{MSE}$|4.51|-|
>
> Applying our method on the baseline introduces slight computational and training time costs, and this cost can be effectively reduced when training with the Distributed Data Parallel framework of PyTorch, demonstrating the efficiency of our method.
>
> ***
> **5. Extensive fine-tuning required for different tasks can be computationally expensive and time-consuming.**
>
> Thanks for your comment! We would like to clarify that our method is evaluated on various datasets under zero-shot and close-set settings using **one unified model**, avoiding the need for multiple fine-tuning processes. Our method can also be applied to parameter-efficient training frameworks, such as adapters, which exhibit impressive training efficiency.  The details of this part please refer to the response of the second question. We will actively explore this direction in future work.
>
> ***
> **6. How does the performance of MoTE vary with different numbers of temporal layers and experts? Are there optimal configurations for specific tasks?**
>
> Thanks for your comment! We have ablated the number of temporal layers and experts in **Figure 1 (b)** and **Table 2 (b)** in the main paper. Please refer to the figure and table in the main paper for the results.
>
> According to the results,  we observe that 4 temporal experts per layer are sufficient to learn various knowledge for both close-set and zero-shot tasks. More experts do not lead to further performance gains. As for the number of the temporal layers, our method consistently outperforms the vanilla Transformer layer and 6-layer MoTE achieves the best trade-off between close-set and zero-shot performance. Therefore, we conclude that the 6 layers MoTE with 4 experts per layer are generally the optimal configuration for various tasks.
>
> ***
> **7. What measures can be taken to reduce the computational overhead introduced by the additional components?**
>
> Thanks for your thoughtful question! Our work is applied to on the standard Transformer structure in the paper. Replacing it with some efficient structure (e.g. flash attention) can effectively reduce the computational overhead of the additional modules. Besides, applying our method to efficient structures (e.g. adapters) can also reduce the computational cost of the additional components.
>
> ***
> **8. Relevant works**
>
> Thanks for your suggestion! We find the recommended papers relevant to our work. We'll cite and discuss them in the revised manuscript.
>
> ***
> We sincerely hope that this response will address the reviewers' concerns. We will supply the above responses in the revised manuscript.

---

> > ### Comment · Area_Chair_BPZY · 2024-08-10
> > **Have a discussion**
> >
> > This paper receives mixed reviews. The authors have provided a detailed response. Please give your reply and check whether there is still unclear point for authors to clarify.

---

### Official Review · Reviewer_7UgE · 2024-07-11

**Soundness:** 3
**Presentation:** 3
**Contribution:** 2
**Rating:** 6
**Confidence:** 4

**Summary:**

This paper addresses the issue of Video-Language Models (VLMs), such as CLIP, experiencing reduced generalization performance to unseen categories when learning domain-specific knowledge for video understanding tasks. The authors propose the MoTE framework, which introduces temporal experts and employs a Mixture of Experts (MoE) approach to effectively learn domain-specific knowledge for videos. Additionally, a soft stochastic routing policy is utilized to further enhance the learning efficiency of the experts. To guarantee the discrepancy in knowledge learned by different experts while maintaining a flat loss landscape, the paper incorporates weight merging regularization, which improves the generalization performance of the learned features. Moreover, the paper presents a temporal feature modulation method that leverages the semantic relevance confidence of proxy text features to modulate features.

**Strengths:**

1. The paper introduces the Mixture of Experts (MoE) approach in zero-shot video classification tasks based on Video-Language Models (VLMs). By utilizing weight merging regularization and other methods, the approach ensures effective learning of domain-specific knowledge in videos while maintaining strong model generalization.

2. The study effectively combines temporal modeling of visual content with the MoE approach. During downstream task adaptation, it leverages multi-perspective data bias learning to avoid overfitting, thus enhancing the learning effectiveness of domain-specific knowledge in videos.

3. The paper analyzes model generalization from the perspective of loss landscape flatness. By improving the flatness, weight merging regularization enhances the generalization performance of the learned features.

**Weaknesses:**

1. There is ambiguity in the use of certain symbols within the paper. For example, the symbol L is used to represent both the loss function of CLIP and the number of layers in the Transformer introduced in MoTE. This issue is particularly evident in Equations (4) and (7). The paper should consider adjusting the usage of these symbols to avoid confusion.

2. There seems to be a problem with the calculation in Equation (5). The notation "exp" typically represents the exponential function of e, but this is not clearly explained. According to the equation, the probability of selecting an expert increases with i, which seems to contradict the intended randomness of stochastic. This requires clarification or correction.

3. In the Introduction and Section 3.4, the paper emphasizes the plug-and-play characteristic of the modulation module. However, the subsequent experiments only demonstrate the improvement in model performance without introducing additional training parameters (Play). They do not showcase the flexibility and usability of the module regardless of the upper model structure (Plug). Therefore, it would be beneficial to add experiments validating the plug-and-play effect or adjust the relevant descriptions in the paper.

**Questions:**

1. What is the design basis for the candidate set of the temperature hyperparameter in weight merging? The paper does not provide a reference for the design of this candidate set, nor does it further validate its superiority over continuous space selection schemes in the experimental analysis.

2. What is the connection between the modulation method proposed in Section 3.4 and the paper's overall motivation? The issue of constrained semantic space it addresses does not seem to be related to the MoE method or the maintenance of feature generalization.

3. What is the specific idea behind the trade-off metric mentioned in Section 4.3? Considering the balance between the two, the arithmetic mean does not seem to be a good metric. If a model achieves 100% ZS performance but 0% close-set performance, its trade-off metric result would be the same as if both values were 50%. How is this issue addressed?

**Limitations:**

The paper does not adequately explain the connection of data bias view and MoE in Section 3.2. For reading-friendly, there should be additional descriptions of the relationship between experts and data bias views, and how the MoE approach leverages multiple data bias views to improve model performance.

---

> ### Author Rebuttal · Authors · 2024-08-06
>
> We appreciate the valuable comments and the time you have dedicated to this paper. Please find the responses to your comments below.
> ***
> **1. Ambiguity in the use of certain symbols.**
>
> Thanks for your kind reminder, and sorry for the confusion. We double-checked the usage of all symbols and will correct the mistakes in the revised manuscript.
> ***
> **2. Missing the notation of the operation exp(). The probability of selecting an expert increases with i, which seems to contradict the intended randomness of stochastic.**
>
> Thanks for your comments! The randomness of expert selection can be described and realized under different probability distributions. For example, the vanilla stochastic routing algorithm follows the uniform probability distribution, while ours follows the multinomial probability distribution. Compared to the uniform distribution, the multinomial probability distribution is less random but more controllable (by assigning different activation probabilities to each expert as in Equation (5)). This allows experts with a greater index to be activated more likely and therefore receive a larger volume of data during training. As a result, each expert can learn knowledge with different degrees of generalization and specialization.
>
> We will include the explanation of the notation exp() and the above discussion in the revised manuscript.
>
> ***
> **3. It would be beneficial to add experiments validating the plug-and-play effect or adjust the relevant descriptions in the paper.**
>
> Thanks for your thoughtful suggestion! We agree that the plug effect should be further validated. Actually, Temporal Feature Modulation can be applied when the temporal module is separate from the CLIP encoder. Considering this constraint, we decided to adjust the relevant descriptions to make them more accurate and rigorous:
>
> Original:
>
> "We propose a plug-and-play module that measures the confidence of temporal features by means of CLIP’s text space."
>
> Revised:
>
> "We propose a test-time adaptation strategy for networks where the temporal module is separated from the CLIP encoder, to measure the confidence of temporal features by means of CLIP’s text space."
> ***
> **4. What is the design basis for the candidate set of the temperature hyperparameter in weight merging? Missing comparison with the continuous space selection schemes.**
>
> Thanks for your valuable question! The candidate set of the temperature hyperparameter {$±2^n · β$} $_{n=0}^4$ ∪ {∞} is designed by ourselves. We set the candidate temperature parameters to grow exponentially so that we can explore a larger range in the weight space (i.e. the coefficients calculated from Equation 7 with different temperatures vary more). {∞} is a special case where all the experts are averagely merged according to Equation 7.
>
> The comparison with the continuous space selection schemes is presented below. We implement two continuous space selection schemes. (1) Sampling from a continuous standard normal distribution (mean=0, variance=1). (2) Sampling from a continuous uniform distribution. The continuous space sampling strategy results in a notable performance degradation.
>
> |Type|K400 (close-set)|UCF|K600|
> |-|:-:|:-:|:-:|
> |Discrete set (default)|86.8|87.5|78.9|
> |Continuous normal dist.|86.5|87.1|77.4|
> |Continuous uniform dist.|86.0|85.9|77.9|
>
> ***
> **5. The connection between the modulation method, the issue of constrained semantic space, and the paper's overall motivation.**
>
> Thanks for your constructive suggestion! This part is correlated with the second objective of our paper: *How can generalization and specialization coexist in one unified model (line 45-46)?* The additional parameters can model the temporal information quite well when dealing with the fine-tuning categories. However, since **the semantic space is constrained** during fine-tuning, the additional parameters may not model the temporal information properly when facing unknown categories. This concern grows as the test categories are less semantically correlated with the fine-tuning categories. Thus, we propose the **modulation module** that measures the confidence of the temporal feature by the semantic association of proxy fine-tuning and text categories. This strategy allows us to **improve the model generalization while keeping its specialization performance**.
>
> We will refine the above discussion and add it to the revised manuscript.
>
> ***
> **6. Improper trade-off metric.**
>
> Thanks for your constructive question! We agree that the reviewers' concern is thoughtful and reasonable. We change the trade-off metric to the harmonic mean of the zero-shot performance A and close-set performance B, calculated as $\frac{2AB}{A+B}$. This metric is not sensitive to extreme values and provides a more accurate indication of the overall performance.
>
> ***
> **7. The relationship between experts and data bias views, and how the MoE approach leverages multiple data bias views to improve model performance.**
>
> Thanks for your suggestion, and sorry for the confusion. ''Data bias views'' indicates the diverse knowledge learned from various optimization trajectories. We set different optimization trajectories for each expert and construct a more generalized model using distinct knowledge learned across experts. We will add the above description to the revised manuscript.
> ***
> We sincerely hope that this response will address the reviewers' concerns. We will supply the above responses in the revised manuscript.

---

> > ### Comment · Area_Chair_BPZY · 2024-08-10
> > **Have a discussion**
> >
> > This paper receives mixed reviews. The authors have provided a detailed response. Please give your reply and check whether there is still unclear point for authors to clarify.

---

> > > ### Comment · Reviewer_7UgE · 2024-08-13
> > > **Reponse to Rebuttal**
> > >
> > > The authors have addressed my concerns, I would like to raise the score to Weak Accept.
> > >
> > > Thank you.

---

### Official Review · Reviewer_s5Zs · 2024-07-12

**Soundness:** 3
**Presentation:** 3
**Contribution:** 3
**Rating:** 6
**Confidence:** 4

**Summary:**

This paper introduces MoTE (Mixture-of-Temporal-Experts) to improve the generalization and specialization capabilities of visual-language models (VLMs) when adapting to video tasks. MoTE addresses two main questions: how to enhance the generalization of additional parameters during fine-tuning, and how to balance generalization and specialization in a unified model. The approach uses multiple feedforward network (FFN) experts in each Transformer layer to capture various data bias views, improving generalization. A routing algorithm based on multinomial distribution maximizes knowledge diversity among experts, while Weight Merging Regularization effectively combines generalized and specialized knowledge in the final model.

To further improve generalization at test time, MoTE incorporates a Temporal Feature Modulation module. Notably, the approach maintains the same computational cost and final structure as conventional methods. The paper contributes to the field by offering a new perspective on enhancing parameter generalization and balancing it with specialization in the context of adapting VLMs to video tasks. Extensive experiments demonstrate that MoTE achieves an optimal trade-off between zero-shot and close-set performance, with thorough ablation studies showing the scalability and effectiveness of the proposed method.

**Strengths:**

- The manuscript is well-written and easy to follow.

- It is interesting to observe that the introduction of a mixture of experts can enhance the balance between acquiring generalizable knowledge and learning video-specific features. The motivation is intuitive, and the extensive experiments effectively validate the method’s efficacy.

- The design of weight merging regularization and temporal feature modulation harmonizes the pursuit of the two learning objectives. The temporal feature modulation is particularly noteworthy, as it takes into account the categorical relationships between the training and test sets to inform the integration of features.

**Weaknesses:**

- The primary motivation for this study stems from two objectives: (1) mitigating the catastrophic forgetting that emerges with the integration of trainable parameters, and (2) striking a balance between generalizable knowledge and video-specific learning within one single model. However, these objectives bear considerable resemblance to the work presented in the paper FROSTER (ICLR 2024), which has not been discussed by the authors. While I acknowledge that the current paper and FROSTER employ distinct methodologies to address these issues, their close relevance necessitates a thorough discussion and a direct performance comparison.

- According to the description in the paper, the baseline model utilizes a clip encoder equipped with several temporal transformer layers. This leads me to question whether the model can be effectively integrated with alternative network architectures, such as adapter-based networks, X-CLIP, and ST-adapter, particularly given their noted efficiency in training.

- I would also request that the authors provide details regarding the additional computational and training time costs associated with implementing their method in conjunction with the baseline model.

- I believe it would be beneficial to delve deeper into the specific types of actions that each expert excels at recognizing. Providing a more detailed analysis in this area would enhance our comprehension of the distinct roles played by various experts, as well as the unique temporal knowledge they contribute in comparison to one another.

[1] FROSTER: Frozen CLIP Is A Strong Teacher for Open-Vocabulary Action Recognition. ICLR 2024.

**Questions:**

Please refer to the weaknesses.

**Limitations:**

The authors have not sufficiently addressed the limitations of their methodology, as it has been applied exclusively to a specific type of adapted network without demonstrating broader applicability. It would be advantageous to see an exploration of the method’s versatility across different network architectures.

---

> ### Author Rebuttal · Authors · 2024-08-06
>
> We appreciate the valuable comments and the time you have dedicated to this paper. Please find the responses to your comments below.
> ***
> **1. Discussion on FROSTER.**
>
> Thanks for your kind reminder! Our motivation does resemble FROSTER's in some way but differs in the following aspects.
>
> *Motivation:*
>
> * FROSTER aims to mitigate the catastrophic forgetting caused by **fine-tuning the model**, while our method focuses on eliminating the catastrophic forgetting caused by **the integration of the additional trainable parameters**. The two motivations stem from different observations and perspectives.
>
> * FROSTER aims to strike a balance between generalized knowledge and video-specific learning **specifically for zero-shot video recognition**, while our work pursues this objective **in more general and various task settings** (close-set, zero-shot, and few-shot recognitions). Since the close-set setting requires much more video-specialized knowledge than the zero-shot setting, our objective is more challenging and practical.
>
> *Contribution:*
>
> * Our method and FROSTER address the objective with different methodologies. FROSTER focuses on knowledge distillation from the frozen CLIP, while our method concentrates on the model structure design and loss landscape regularization. We offer distinct contributions to the community.
>
> We will add the above discussion and performance comparison in the revised manuscript.
> ***
> **2. Whether the model can be effectively integrated with alternative network architectures, such as adapter-based networks?**
>
> Thanks for your constructive suggestion! To further demonstrate the generality and scalability of MoTE, we apply our method to ST-Adapter [1]. For simplicity, we remove the depth-wise Conv3D operation in our adapter. In this case, our baseline is slightly lower than ST-Adapter, but it doesn't affect justifying the effectiveness of our method. The results are shown in the table below.
>
> |Method|K400 (close-set)|UCF|K600|
> |-|:-:|:-:|:-:|
> |ST-Adapter|81.1|77.3|61.1|
> |ST-Adapter+MoTE|81.2|79.4|66.7|
>
> Our method delivers a notable generalization performance improvement while maintaining the close-set result, indicating the effectiveness of MoTE applied to alternative networks.
>
> [1] ST-Adapter: Parameter-Efficient Image-to-Video Transfer Learning. NeurIPS 2022.
> ***
> **3. Additional computational and training time costs associated with implementing the method in conjunction with the baseline model.**
>
> Thanks for your comment! We show the training time costs in Table 8 of the supplementary material. Below, we include the table for your convenience and additionally add GFLOPs for reference. GPU days are calculated by the number of GPUs multiplied by the training time in days.
>
> |Method|GPU-days|GFLOPs|
> |-|:-:|:-:|
> |Baseline|4.35|649|
> |+MoTE|4.35|+0.34|
> |+$L_{WMR}$|4.50|+0.34|
> |+$L_{MSE}$|4.51|-|
>
> As shown in the table, applying our method on the baseline introduces slight computational and training time costs, illustrating the efficiency of our method.
> ***
> **4. More detailed analysis of the specific action types that each expert excels at recognizing.**
>
> Thanks for your constructive suggestion! We perform a category-wise performance analysis of the merged experts and each individual expert. We visualize the Top-1 classification accuracy for each video category, which is sampled from the UCF-101 dataset. The experiment is conducted on UCF-101 under the zero-shot setting. **The visualization is provided in the attached global response PDF.**
>
> We find that different experts exhibit distinct performance for each video category. This is because diverse generalization knowledge is learned across experts, as we demonstrate in the paper. The merged experts benefit from the aggregation of knowledge across experts, especially for some video categories where temporal information is particularly needed, for example, 'Body Weight Squats', 'Handstand Pushups', and 'Wall Pushups'. This suggests that each expert can learn various temporal patterns for the same category. For more visualizations, please refer to Figure 5 and Figure 6 in the supplementary material, which shows more intuitive visualizations.
>
> ***
> We sincerely hope that this response will address the reviewers' concerns. We will supply the above responses in the revised manuscript.

---

> > ### Comment · Area_Chair_BPZY · 2024-08-10
> > **Have a discussion**
> >
> > This paper receives mixed reviews. The authors have provided a detailed response. Please give your reply and check whether there is still unclear point for authors to clarify.

---

> > > ### Comment · Area_Chair_BPZY · 2024-08-10
> > > **Performance comparison with  FROSTER**
> > >
> > > After reading the reviews and the author rebuttal, the AC thinks the authors should provide more detailed performance comparison with FROSTER as requested by the reviewers. The paper shares very similar motivation to FPOSTER, while this paper missed citation and discussion in the original submission. So please make a detailed comparison here, and it will influence the final decision.

---

> > > > ### Author Response · Authors · 2024-08-11
> > > > **Response to the performance comparison with FROSTER.**
> > > >
> > > > Thanks for AC's thoughtful suggestion! We provide a detailed performance comparison with FROSTER in the table below.
> > > >
> > > > *Performance comparison:*
> > > >
> > > > |Method|K400 (close-set)|UCF|HMDB|K600|Trade-off|
> > > > |-|:-:|:-:|:-:|:-:|:-:|
> > > > |FROSTER|78.9|**84.8**|54.8|**74.8**|74.0|
> > > > |MoTE|**81.8**|83.4|**55.8**|70.2|**74.9**|
> > > >
> > > > As shown in the table, FROSTER is a strong zero-shot action recognition model but performs less well on close-set action recognition. The reason lies in that (1) The distillation supervision from the Frozen CLIP limits the model's ability to learn video-specialized knowledge. (2) FROSTER applies the weights ensemble to average the models learned in different epochs, which improves the generalization at the cost of close-set performance. These results suggest that FROSTER doesn't really achieve a sound balance between the close-set and zero-shot tasks. On the contrary, our method demonstrates remarkable performances on both close-set and zero-shot tasks, striking a better balance between the two aspects (higher Trade-off score).
> > > >
> > > > Besides, we would like to note that FROSTER's superior K600 performance partly comes from their additional fine-tuning of the text encoder and the use of GPT rephrased action descriptions. When removing these modifications, our method achieves competitive K600 results with FROSTER (70.2% v.s. 71.1%).
> > > >
> > > > *More motivation clarification:*
> > > >
> > > > * FROSTER's motivation rests on their observation that the model's zero-shot performance declines after fine-tuning. Differently, our motivation is based on the observation that the **zero-shot generalization diminishes with the increase in the scale of additional specialized parameters**. The main purpose of our work is to explore how to manage the generalization/specialization trade-off posed by adding additional parameters in transfer learning, which is not discussed in FROSTER.
> > > > * The balance objective pursued by FROSTER is designed for zero-shot action recognition. Since the model's generalization and specialization capabilities are both measured by only zero-shot results, it is difficult to conclude whether the balance is achieved. On the contrary, in our work, generalization and specialization capabilities are separately measured by zero-shot and close-set results, allowing us to further explore their properties. In this case, our balance objective is more emphasized on **how to manage the conflicting nature of generalization and specialization during model training**.

---

> > > > > ### Comment · Reviewer_s5Zs · 2024-08-11
> > > > > **Response to the Response to the performance comparison with FROSTER**
> > > > >
> > > > > Thanks for your additional experiments, which look great. I strongly suggest the authors include the discussion and experiments in your revision, that would make the paper more convincing.

---

> ### Comment · Reviewer_s5Zs · 2024-08-11
> **Response to rebuttal**
>
> Thank you to the authors for their comprehensive rebuttal, which has addressed many of my concerns.
>
> However, I remain unconvinced by the explanation provided for the first point of contention.
>
> >FROSTER aims to mitigate the catastrophic forgetting caused by **fine-tuning the model**, while our method focuses on eliminating the catastrophic forgetting caused by the integration of the **additional trainable parameters**. The two motivations stem from different observations and perspectives.
>
> Nevertheless, upon examining the FROSTER paper, it becomes evident that the term **fine-tuning the model** encompasses approaches that incorporate **additional trainable parameters** within CLIP. Notably, in Figure 1 and Table 3 of the FROSTER paper, experiments with adapter-based methods such as AIM and ST-adapter are presented, which clearly involve the addition of extra parameters.
>
> Given this context, the authors’ response does not sufficiently address the overlap between the two methodologies. I would appreciate further clarification on this matter.
>
> Besides, I would like to know how the authors initialize the parameters of MOTE.
>
> Thanks!

---

> > ### Author Response · Authors · 2024-08-12
> > **Further clarification on FROSTER and the initialization of MoTE**
> >
> > **1. Further clarification on FROSTER:**
> >
> > Thanks for pointing this out. After carefully reading the FROSTER paper, we find that the catastrophic forgetting problem in FROSTER is caused by the task-specific learning steers the learned features diverging too far from the frozen CLIP. This divergence comes from optimizing parameters through gradient descent, including both CLIP and additional trainable parameters.  Differently, in our work, we believe that **the main cause of catastrophic forgetting is the overfitting of additional parameters rather than the variations in CLIP parameters**. This can be evidenced by our observation that the diminishment in the model's generalization is closely related to the scale of the additional parameters, regardless of whether the CLIP parameters are tuned. Therefore, our work focuses on how to improve the generalization **specifically for additional parameters** while FROSTER aims to enhance the generalization of the **overall feature** by ensuring the learned features do not diverge too far from the frozen CLIP. Our motivation indeed bears some resemblance to FROSTER's in addressing catastrophic forgetting but differs in (1) the observations that lead to the motivation (2) the perspective of increasing model generalization.
> >
> > From the technical perspective, our work and FROSTER provide distinct contributions to the community by proposing different methodologies. Note that FROSTER can also potentially improve the generalization of additional parameters through knowledge distillation, but our method presents a more explicit way to achieve this goal.
> >
> >
> > **2.The initialization of MoTE**
> >
> > The parameters of projection matrices (nn.Linear) are initialized with values drawn from the normal distribution (mean=0, std=0.02). Each projection matrix has different initial values to ensure different optimization trajectories. All bias terms are initialized as zero.
> >
> > **3. Response to the performance comparison with FROSTER**
> >
> > Thank you! We will include the discussion of FROSTER and experiment results in our revised manuscript.

---

> > > ### Comment · Reviewer_s5Zs · 2024-08-12
> > > **Response to the further clarification**
> > >
> > > Thanks for your feedback.
> > >
> > > 1. I am generally satisfied with the discussion of the comparison between FROSTER and MOTE.
> > >
> > > 2. The authors only introduce how to initialize linear layer parameters. However, as shown in Fig. 2 of the main paper, MOTE also contains self-attention layers. Then, how do you initialize them?
> > >
> > > 3. I'm glad to see that.

---

> > > > ### Author Response · Authors · 2024-08-12
> > > > **The initialization of self-attention layers**
> > > >
> > > > Thanks for your encouragement, and sorry for missing the self-attention layer. The initialization of the self-attention layer is consistent with that mentioned above. All projection matrices in the self-attention layer are initialized with values drawn from the normal distribution (mean=0, std=0.02).  All bias terms are initialized as zero. As for the LayerNorm, the $\gamma$ (weight) is initialized as one and the $\beta$ (bias) is initialized as zero.

---

> > > > > ### Comment · Reviewer_s5Zs · 2024-08-12
> > > > > **Reponse to the authors**
> > > > >
> > > > > I have no further questions. Please ensure that you improve your paper based on the discussion.
> > > > >
> > > > > The authors solved all my concerns, I have raised my score to 6.
> > > > >
> > > > > Thank you.

---

> > > > > > ### Author Response · Authors · 2024-08-12
> > > > > >
> > > > > > Thank you for all the suggestions, and we'll certainly include all the discussion in the revised manuscript!

---

### Official Review · Reviewer_KXzA · 2024-07-14

**Soundness:** 3
**Presentation:** 3
**Contribution:** 3
**Rating:** 6
**Confidence:** 4

**Summary:**

To preserve the generalization ability of the model trained on general visual-language model (VLM) with task-specific data, while boost the performance on specific task, this paper propose a new framework and training strategy to learn a unified model with specific performance and generalization ability. Three techniques are introduced. Mixture temporal experts to avoid overfitting on the task-specific data. A weight merging regularization to enlarge the loss flat region such that optimization on generalization ability will not introduce perturbation that drops the close-set performance. A temporal feature modulation to reuse the feature of VLM model when the target category label is not fitted during task-specific finetuning. The proposed method is evaluated on four benchmark datasets. K400 for close-set finetuning and UCF-101, HMDB-51and K600 for zero-shot evaluation.

**Strengths:**

1.	To train a model with both task-specific performance and zero-shot generalization ability is a interesting topic, and it is less explored in the community.
2.	The proposed method achieves competitive performance compared with the similar methods.
3.	Balancing between the zero-shot and the task-specific ability is always hard to handle. Considering the wide application of general VLM, this method bears practical value in the industry.

**Weaknesses:**

1.	The experimental setting may hide the weakness of the proposed method. The method is only trained on K400 and evaluated its zero-shot ability on UCF-101, HMDB-51and K600. Considering K400 is already a large-scale dataset, the MoTE may still have good performance on UCF-101 and HMDB-51. Besides, K600 is an extension of K400, therefore they may have similar data distribution. It would be great to also finetune the model on small-scale dataset and evaluated generalization ability on large-scale dataset, for example, train the model on UCF-101 and evaluate it on K400.
2.	A simple solution to handle the zero-shot / task-specific balancing issue is to use a finetuned model such as Text4Vis for specific task and to use its temporally mean-pooled clip feature when facing out-of-distribution task. This baseline is missing in the comparison. If the performance of this baseline is acceptable, is it really necessary to train a unified model with such much cost?

**Questions:**

1.	The second question in the weakness section
2.	The VLM Clip is actually trained on noisy data, and there are also VLM trained with selected data to boost its cross-modality alignment [1]. Therefore, the selection of K in line189 may have influence on the final performance. Besides, for different text-query, the influence of noisy data is diffrerent, and one fixed K may not be optimal. Is there any solution for this issue. Is the selection of K have large influence on the performance?
[1]  Bulat, Adrian, Yassine Ouali, and Georgios Tzimiropoulos. "FFF: Fixing Flawed Foundations in contrastive pre-training results in very strong Vision-Language models." Proceedings of the IEEE/CVF Conference on Computer Vision and Pattern Recognition. 2024.

**Limitations:**

The limitation has been discussed in the suplemental material.

---

> ### Author Rebuttal · Authors · 2024-08-06
>
> We appreciate the valuable comments and the time you have dedicated to this paper. Please find the responses to your comments below.
> ***
> **1. The experimental setting may hide the weakness of the proposed method. It would be great to also fine-tune the model on the small-scale UCF-101 and evaluate it on K400.**
>
> Thanks for your thoughtful and reasonable suggestion! Our experimental setting follows previous works (Methods listed in Table 3) in fine-tuning on large-scale K400 and then evaluating zero-shot performance on relatively small downstream datasets. Following your suggestion, we train the ViT-L/14 network on UCF-101 and evaluate it on K400, results are shown in the table below.
>
> |Method|UCF (close-set)|K400 (zero-shot)|
> |-|:-:|:-:|
> |Raw CLIP|80.9|59.2|
> |Baseline|95.6|56.6|
> |MoTE|96.1|66.3|
>
> MoTE yields a 0.5 improvement over the Baseline on UCF-101, while zero-shot performance on K400 significantly outperforms both Raw CLIP and Baseline. This demonstrates the applicability of MoTE to small-scale datasets and its ability to learn generalized knowledge from limited data.
> ***
> **2. Comparison with the baseline variant. Is it really necessary to train a unified model with such much cost?**
>
> We adopt the recommended baseline and present the results in the table below. Our method still shows notable advantages in zero-shot performance.
>
> |Method|K400 (close-set)|UCF|HMDB|K600|
> |-|:-:|:-:|:-:|:-:|
> |Baseline Variant|86.7|87.1|58.2|77.4|
> |MoTE|86.8|88.7|61.4|79.0|
>
> We would like to claim that the most significant value of our work is its ability to simultaneously introduce new specialized knowledge while keeping original generalized knowledge, rather than solely achieve the best performance on a particular dataset. Although the recommended baseline performs fairly on close-set and zero-shot tasks, it does not really achieve our goal. This can be evidenced when facing a middle-ground task between close-set and zero-shot tasks, for example, the few-shot task. It requires both specialization and generalization capability to rapidly adapt using limited samples. We present the results of the few-shot task in the table below. Our method significantly outperforms the recommended baseline.
>
> |Method|UCF-2 shot|UCF-4 shot|UCF-8 shot|UCF-16 shot|HMDB-2 shot|HMDB-4 shot|HMDB-8 shot|HMDB-16 shot|
> |-|:-:|:-:|:-:|:-:|:-:|:-:|:-:|:-:|
> |Baseline (clip feature)|78.6|81.6|82.3|83.0|53.7|54.2|55.2|55.8|
> |Baseline (temporal feature)|86.4|88.9|90.6|91.2|55.9|61.2|65.4|66.3|
> |MoTE|88.8|91.0|92.3|93.6|61.3|63.9|67.2|68.2|
>
> From the perspective of model structure, the recommended baseline can only be applied when the additional parameters are separated from the CLIP encoder. Our proposed method can be applied to various forms of additional parameters (e.g. adapter) that are integrated into the CLIP encoder, demonstrating the versatility of our method. The details of this part please refer to the response of the second question to the reviewer s5Zs. Moreover, in real-world applications, the data distribution shift is often hard to identify. The recommended baseline faces the choice of whether to use the temporal feature when facing a moderate or unknown distribution shift, which limits its practicality. This further demonstrates the necessity of training a unified model.
>
> ***
> **3. For different text-query, the influence of noisy data is diffrerent, and one fixed K may not be optimal. Is there any solution for this issue. Is the selection of K have large influence on the performance?**
>
> Thanks for your interesting question! We agree with your opinion and conduct an experiment. In this experiment, we employ the K-means algorithm to cluster the fine-tuning and test category text features and compute the semantic association by retrieving the most similar cluster-centered feature. In this case, the retrieved cluster-centered feature may represent different numbers of data points. We hope this strategy can better represent the semantic associations between fine-tuning and test categories. We also test the effect of the number of K. The results are shown in the table below (TFM indicates the Temporal Feature Modulation).
>
> |Method|TFM|UCF|K600|
> |-|:-:|:-:|:-:|
> |MoTE|-|87.5|78.9|
> |MoTE (clustering)|√|88.8|78.9|
> |MoTE (K=1)|√|88.9|78.9|
> |MoTE (K=5)(default)|√|88.7|79.0|
> |MoTE (K=10)|√|88.5|78.7|
>
> We find that applying the clustering algorithm does not lead to performance improvement. The reason lies in that using the semantic association to measure the confidence of the temporal feature is actually a rough estimation process. Since this process is inherently somewhat noisy, better-selected data may not largely affect it. This also explains why the selection of the K-value does not have a significant influence on the performance. However, we still observe performance degradation when K is too large, indicating that the noise introduced from the selection of the K dose has a negative influence on the performance.
> ***
> We sincerely hope that this response will address the reviewers' concerns. We will supply the above responses in the revised manuscript.

---

> > ### Comment · Area_Chair_BPZY · 2024-08-10
> > **Have a discussion**
> >
> > This paper receives mixed reviews. The authors have provided a detailed response. Please give your reply and check whether there is still unclear point for authors to clarify.

---

### Author Rebuttal · Authors · 2024-08-06

We are grateful to all reviewers for their valuable and constructive comments. We have carefully considered the points raised by each reviewer and provided comprehensive responses to each question. Besides, we attach an additional PDF file containing a detailed analysis of the category-wise performance across experts.

---

### Decision · Program_Chairs · 2024-09-25

**Decision:**

Accept (poster)

**Comment:**

This paper receives unanimous weak accept recommendations. After a careful check of paper, comments, and rebuttal, the AC agrees with the reviewers and think the authors has well addressed the reviewer concern. The paper introduces a interesting idea to balance the generalization and speicalization ablity during transfer learning of VLM. The results demonstrate the effectiveness of proposed method.Thus, the AC makes a accept recommendation.